# SMILE: A Composite Lexical-Semantic Metric for Question-Answering Evaluation

**Shrikant Kendre**[*]
*Salesforce AI Research*

*skendre@salesforce.com*

**Austin Xu**[*]
*Salesforce AI Research*

*austin.xu@salesforce.com*

**Honglu Zhou**
*Salesforce AI Research*

*honglu.zhou@salesforce.com*

**Michael S. Ryoo**
*Salesforce AI Research*

*mryoo@salesforce.com*

**Shafiq Joty**[†]
*Salesforce AI Research*

*sjoty@salesforce.com*

**Juan Carlos Niebles**[†]
*Salesforce AI Research*

*jniebles@salesforce.com*

**Reviewed on OpenReview:** *https://openreview.net/forum?id=lnpOvuQYih&noteId=lnpOvuQYih*

## Abstract

Traditional evaluation metrics for textual and visual question answering, like ROUGE, ME-TEOR, and Exact Match (EM), focus heavily on n-gram based lexical similarity, often missing the deeper semantic understanding needed for accurate assessment. While measures like BERTScore and MoverScore leverage contextual embeddings to address this limitation, they lack flexibility in balancing sentence-level and keyword-level semantics and ignore lexical similarity, which remains important. Large Language Model (LLM) based evaluators, though powerful, come with drawbacks like high costs, bias, inconsistency, and hallucinations. To address these issues, we introduce **SMILE**: Semantic Metric Integrating Lexical Exactness, a novel approach that combines sentence-level semantic understanding with keyword-level semantic understanding and easy keyword matching. This composite method balances lexical precision and semantic relevance, offering a comprehensive evaluation. Extensive benchmarks across text, image, and video QA tasks show SMILE is highly correlated with human judgments and computationally lightweight, bridging the gap between lexical and semantic evaluation. Code and evaluation scripts are available at https://github.com/SalesforceAIResearch/smile-metric-qna-eval

## 1 Introduction

Question answering (QA) is an essential task used to measure the progress of language-based models. Across text, image, and video domains, the primary measure of model performance on QA benchmarks is accuracy, which is typically computed via *exact (or easy) match (EM)*: A model response is deemed correct if the ground-truth answer, typically annotated by humans, exactly matches (or can be found within) the model response. As recent models have grown to be more capable language generators, model answers have grown

---

[*]Equal contribution.
[†]Equal senior contribution.

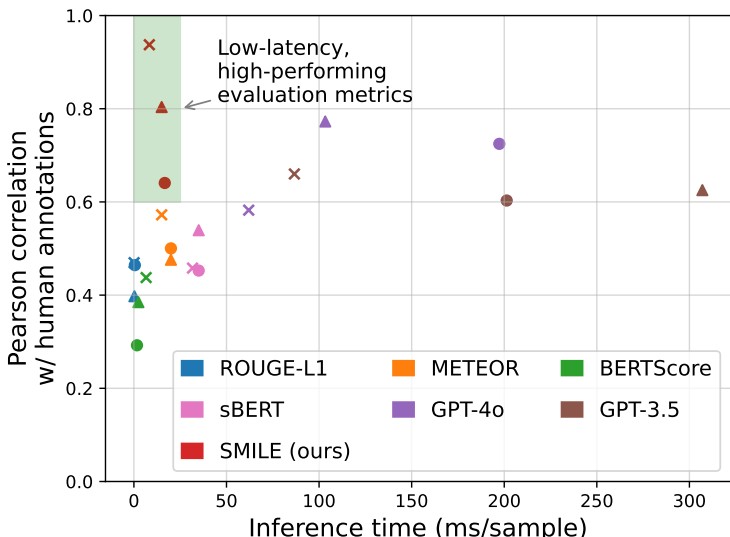

Figure 1: **SMILE offers high-performance, low-latency QA evaluation**, breaking the trade-off between cost and performance. Performance is averaged across human-annotated samples from benchmarks for natural language (×), image (△), and video (∘) domains.

more nuanced, making EM overly stringent (Wang et al., 2023a). A reasonable attempt to mitigate these issues is to employ N-gram based metrics typically used for text generation evaluation, such as ROUGE (Lin, 2004) or METEOR (Banerjee & Lavie, 2005), or embedding-based metrics like BERTScore (Zhang et al., 2019), MoverScore (Zhao et al., 2019), and BLEURT (Sellam et al., 2020), to assess similarity between the predicted response and the ground-truth (Rajpurkar et al., 2016; Bajaj et al., 2016; Dunn et al., 2017; Kočiský et al., 2018; Yang et al., 2018a). While such metrics capture high-level similarity between the model response and ground-truth, they may miss fine-grained details crucial to answer correctness (e.g., "The cat is *on* a chair" vs. "The cat is *under* a chair") that result in lower correlation with human judgments (Mañas et al., 2024).

Concurrently, due to their strong language comprehension abilities, large language models (LLMs) have been deployed as automatic evaluators for text generation. This approach, broadly known as *LLM-as-judge*, functions by either prompting more capable LLMs, like GPT-4o, or finetuning smaller LLMs specifically for evaluation. LLM-as-judge is appealing as LLMs can adapt to different evaluation criteria and generate explanations. Consequently, recent methods and benchmarks (Jacovi et al., 2025; Wang et al., 2024a) now employ judges as evaluators in QA settings.

However, using judge models for evaluation increases *costs*. For practitioners and developers with limited resources, repeatedly querying pay-per-use API models to evaluate large datasets (5K+ samples) or dedicating limited compute to hosting an evaluation server can be impractical for rapid development, which may drive them to use lesser but faster metrics. Beyond resource demands, generative evaluators also exhibit relatively high latency (see Figure 1) and are susceptible to hallucinations, as we qualitatively show in Section 3.

This work revisits embedding-based approaches for automatic QA evaluation and introduces Semantic Metric Integrating Lexical Exactness (SMILE), a lightweight yet high-performing framework for grading QA tasks. SMILE aims to retain the efficiency of embedding-based evaluators while addressing their limitations, such as lack of fine-grained response understanding. To do so, SMILE comprises two subscores: A semantic subscore to assess overall response content, and a keyword subscore to reward lexical exactness. Overall, SMILE offers a best-of-both-worlds evaluation solution: As Figure 1 shows, it correlates with human annotators as strongly as GPT-4o. Additionally, like other embedding-based metrics, SMILE core components can be precomputed for fast lookup, resulting in a 9x speedup compared to LLM-as-judge API queries. SMILE's lightweight design allows it to run on CPU during evaluation, requiring minimal GPU VRAM to perform

a *one time* evaluation dataset preprocessing step. Additionally, SMILE offers interpretability through its composite structure with two subscores. While SMILE can be applied in many settings, our study focuses on factoid QA tasks. Our contributions are:

**(1)** We revisit the promise of embedding-based automatic evaluation metrics for QA tasks and propose SMILE, which utilizes both sentence and keyword level similarity scores to evaluate based on holistic and fine-grained content.
**(2)** We construct a 225-sample human annotated test set from nine (text, image, video) QA datasets, labeled by 4 domain experts. This set is used to benchmark SMILE and other metrics based on their correlation with human judgments.
**(3)** We demonstrate that SMILE can serve as a lightweight drop-in replacement for more powerful LLM-as-judge models across modalities.
**(4)** Extensive ablation studies demonstrate the necessity of each of SMILE's components.

In all, our experiments demonstrate that SMILE is a *lightweight* yet *high-performing* automatic evaluation metric for QA settings. Code and evaluation scripts are available at `https://github.com/SalesforceAIResearch/smile-metric-qna-eval`

## 2 Background and related work

This work addresses three QA modalities: Natural language QA (NLQA), visual QA (VQA), and video QA (VidQA). Across these, a model $f$ receives an input $(q, c)$ and generates a textual answer $y = f(q; c)$. The input consists of a question $q$ and context $c$, where $c$ is text for NLQA, an image for VQA, and a video for VidQA. The task of the model is to produce a natural language answer $y$ to the question $q$ based on the given context $c$.

For model evaluation, we adopt a *reference-based* protocol, assuming a human-annotated ground-truth answer $y^\star$ is available for each input $(q, c)$. Given a model response $y$, the goal is to determine its correctness. Specifically, we aim to design an evaluator $j$ that produces an evaluation score $s = j(y, y^\star)$ based on $y$ and $y^\star$. This *source-free* setup, where evaluation occurs without access to $(q, c)$, aligns with the original exact match (EM) evaluation setup. Source-free evaluation may represent an *easier* evaluation setting, as prior work indicates LLM-based evaluators struggle when context $c$ is included (Xu et al., 2025).

For practitioners prioritizing accuracy, the evaluation score $s$ can be a binary correct/incorrect label. For detailed failure analysis, a finer-grained score (e.g. 0-5 scale) may be preferred. Regardless of the format, the score should be convertible to a binary label, typically via a straightforward threshold (e.g., scores $\leq 3$ as incorrect, scores $\geq 4$ as correct for a 0-5 scale) (Maaz et al., 2024; Wang et al., 2024a). We now review existing metrics and evaluators.

**Text generation metrics.** QA benchmarks have typically used EM or ROUGE (Lin, 2004) to assess model outputs, e.g., (Yang et al., 2018b; Rajpurkar et al., 2016; Dunn et al., 2017; Fan et al., 2019). As model responses grew more nuanced, n-gram metrics such as BLEU (Papineni et al., 2002) and METEOR (Banerjee & Lavie, 2005) were adopted in QA settings (Bajaj et al., 2016). Despite better correlation with human annotations than EM, n-gram metrics have been shown to be insufficient for modern QA tasks (Chen et al., 2019).

**Embedding-based metrics.** Embedding models, like BERT (Devlin et al., 2019) and finetuned variants, e.g., (Reimers & Gurevych, 2019; Gao et al., 2021), are trained to measure semantic similarity. As such, embedding-based metrics are a natural step in overcoming the limitations of EM and n-gram-based metrics (Chen et al., 2019), with notable methods like BERTScore (Zhang et al., 2019), BARTScore (Yuan et al., 2021), BLEURT (Sellam et al., 2020) being used as evaluators in benchmarks (Ao et al., 2024). Recent work (Bulian et al., 2022; Risch et al., 2021; Lee et al., 2020) developed metrics specifically for the QA setting.

**LLM-as-judge evaluators.** The LLM-as-judge paradigm initially utilized frontier LLMs for automatic evaluation (Wang et al., 2023b; Liu et al., 2023c; Fu et al., 2024; Chiang & Lee, 2023). However, biases in prompted evaluators hold were soon identified (Panickssery et al., 2024; Wang et al., 2023c; Park et al.,

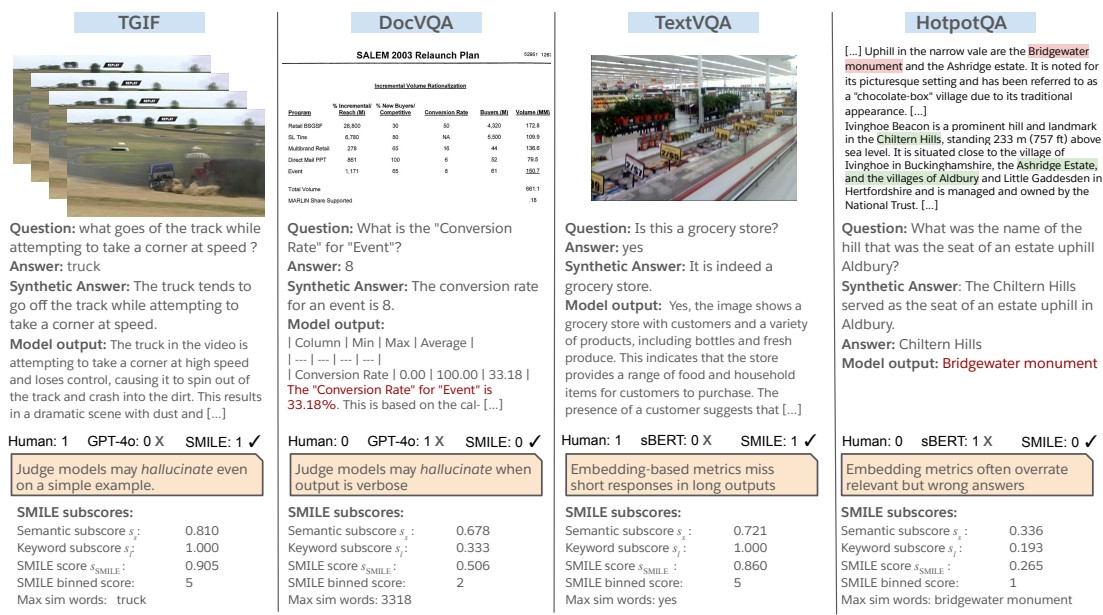

Figure 2: **Example failure cases for existing methods**. *Columns 1-2* illustrate LLM-as-judge failures: hallucination even on simple verification (column 1), and incorrect yet concrete responses (column 2). *Columns 3-4* illustrate embedding-based model failures with lengthy (column 3) and relevant but incorrect (column 4) responses. SMILE correctly identifies each case and exposes expanded, interpretable subscores (bottom): semantic alignment ($s_s$), keyword/faithfulness ($s_l$), overall score ($s_{\text{SMILE}}$), a binned decision score, and the top max-similarity words, providing transparent evidence for why a response is accepted or rejected. Scores are shown on a 0/1 scale for comparison. [...] denotes omitted content for brevity.

2024), leading to the finetuning of smaller models for evaluation (Kim et al., 2024; Li et al., 2023a; Zheng et al., 2024; Wang et al., 2023d; Shiwen et al., 2024). Recent efforts focus on training for diverse evaluation protocols (Vu et al., 2024; Wang et al., 2024b), such as pairwise, single rating, and binary classification. Applying LLM-as-judges specifically to QA grading is a recent development (Mañas et al., 2024), with many benchmarks (Krishna et al., 2024; Jacovi et al., 2025) and studies (Maaz et al., 2024; Liu et al., 2023b) employing API models.

## 3 When do existing evaluation methods and metrics fail?

Before introducing SMILE, we present a qualitative case study highlighting the failure modes of LLM-as-judge and embedding-based metrics in QA evaluation. Our analysis of 225 human-evaluated model responses reveals a common failure point: verbose or generic model outputs, consistent with Mañas et al. (2024); Luo et al. (2021). By "verbose," we refer to predictions significantly longer than the ground-truth answer, often containing tangential information, hedging language, or hallucinated content that obscures the core answer—cases where keyword matching effectively identifies missing critical terms while semantic similarity alone scores highly due to topical relevance.

Surprisingly, we also find that using a powerful LLM like GPT-4o does not guarantee accurate evaluations (Yan et al., 2025; Thakur et al., 2025). We highlight two representative failure modes in Figure 2 (columns 1-2). GPT-4o was prompted to generate a binary accuracy label, as well a 1-5 score (see Section A for prompt).

For ease of comparison, we convert SMILE scores to a binary accuracy indicator. A primary concern with using LLMs as judges is hallucinations. Figure 2 (column 1) shows that even for relatively simple samples, GPT-4o may hallucinate an incorrect label. Figure 2 (column 2) is an example of *concreteness* bias, a known judge model bias (Park et al., 2024). Here, GPT-4o is tricked by a response that includes concrete artifacts, like the table the model generated, even if it incorrect response.

Seeking an efficient evaluator, we also analyzed failure modes of embedding-based approaches. Semantic similarity metrics like BERTScore exhibited well-known limitations (Zhang et al., 2019). Figure 2 (columns 3-4) highlights two key examples: column 3 shows how overly verbose model responses can easily misled semantic similarity metrics, as much of the output is irrelevant to the simple "yes" response. This can be viewed this as *distributional misalignment* (Agrawal et al., 2022): increasingly high-quality model outputs are often lengthier, contrasting with the typical short answers in factoid QA benchmarks. Conversely, when model responses are short but semantically relevant, these metrics are prone to false positives, as illustrated in Figure 2 (column 4). The limitations of embedding-based and LLM-as-judge methods motivated SMILE-a hallucination-free evaluation metric presented next.

## 4 The SMILE metric

Our analysis in Section 3 pinpointed two critical limitations of embedding-based approaches: (1) a *distributional gap* between verbose model responses and concise ground-truth answers, and (2) a *lack of fine-grained understanding* due to their semantic focus. SMILE directly addresses these issues with two key innovations: (1) Synthetic answer generation to bridge the stylistic distribution gap, and (2) targeted sub-scores capturing both semantic and lexical similarity between model responses and ground-truth.

**Bridging the stylistic distribution gap.** As shown in Section 3, assessing directly based on the ground truth $y^\star$, which is typically short for short-form (factoid) QA (e.g., a single word or short phrase), may be sub-optimal, as model responses tend to be more verbose. Motivated by past work that have used LLMs to perform other kinds of zero-shot distribution alignment (Gao et al., 2023; Xu et al., 2024), we utilize an LLM to generate a synthetic model response from the ground-truth. Our key insight is that for short-form QA tasks, a *lightweight* model (e.g., 3B parameter) can be deployed as a synthetic answer generator $g$. Specifically, the generator $g$ takes as input the original question $q$ and ground truth answer $y^\star$ and outputs a synthetic answer $\tilde{y} = g(y^\star, q)$, which aligns stylistically with model responses, but reflects the ground-truth answer content. As a concrete example from our evaluation setup, for input question "What is the Conversion Rate for Event?" and ground-truth "8", a generated synthetic answer is "The conversion rate of an event is 8". We emphasize that *synthetic answer generation is independent of the model being evaluated and is performed only once, prior to test-time, per evaluation set.* As a result, synthetic answers may be stored and used for any subsequent evaluations.

**Integrating semantic and lexical similarity.** The core idea of SMILE is to measure both *semantic* and *lexical* similarity between the model response and the ground-truth using an embedding model $e$. We calculate a semantic similarity score, which we denote $s_s$, as

$$s_s(y, \tilde{y}; e) = \texttt{sim}(e(y), e(\tilde{y})), \tag{1}$$

where, $\texttt{sim}(x, y) = (1 + \langle x, y \rangle / \|x\|_2 \|y\|_2)/2$, which is a linearly transformed cosine similarity that lies within an interpretable interval of $[0, 1]$. As we show in Section 5.4, generating synthetic answers bridges the stylistic distribution gap between ground-truth answers and model responses enough to make semantic similarity meaningful. However, Section 3 shows that this semantic similarity score alone is insufficient to capture the nuances of evaluation. As a result, we additionally compute a lexical similarity score, which we denote $s_\ell \in [0, 1]$, as

$$s_\ell(y, y^\star; e) = \frac{1}{2} \left( \text{EM}(y, y^\star) + \max_i \left\{ \texttt{sim}\left(e(N_i[y]), e(y^\star)\right)\right\} \right), \tag{2}$$

where $EM(y, y^\star) \in \{0, 1\}$ score between the prediction $y$ and ground-truth $y^\star$ and $N_i[y]$ denotes the $i$-th n-gram of response $y$. In computing $s_\ell$, we take advantage of the fact that $y^\star$ is typically a short phrase to compute two complementary scores. The easy match sub-score $EM$ serves as a preliminary check for lexical

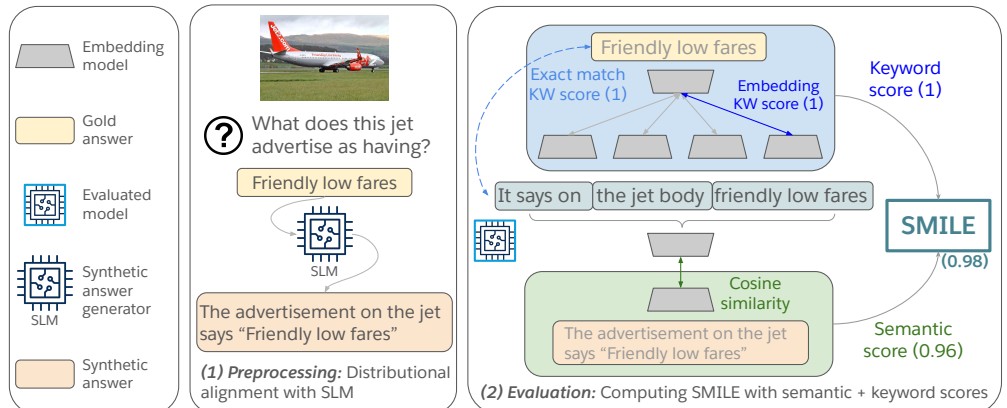

Figure 3: **Overview of SMILE**. SMILE evaluates QA outputs in two steps using a synthetic answer generator and an embedding model. (1) *One-time* preprocessing generates stylistically aligned synthetic answers using a small language model (SLM). (2) SMILE computes two sub-scores: a semantic score based on the synthetic answer, and a keyword subscore that combines exact match with embedding comparisons of model response N-grams to the gold answer. This subscoring balances semantic and lexical evaluation.

answer correctness. However, as noted in prior work (Wang et al., 2023a; Luo et al., 2021), string matching may be too stringent for synonym-like answers (e.g., "cat" vs. "kitten"). As a result, we loosen the necessity for string matches via the maximum n-gram embedding similarity score, which serves as a continuous-valued measure of lexical exactness.

**Evaluation with SMILE.** With our semantic and lexical scores computed, we can now compute the SMILE score, denoted $s_{\texttt{SMILE}} \in [0, 1]$:

$$s_{\texttt{SMILE}}(y, y^{\star}; e, w) = \frac{1}{2}(w \cdot s_s(y, \tilde{y}; e) + (1 - w) \cdot s_{\ell}(y, y^{\star}; e)), \tag{3}$$

where $w \in (0, 1)$ is some user-specified weight to balance the two subscores. This weighting mechanism allows practitioners to express their preferences: Those who are more inclined towards exact match may place higher weight on $s_{\ell}$, whereas those who value higher responses whose meaning is closest with the ground-truth may place a higher weight on $s_s$.

## 4.1 Optimizations for test-time speed-up

SMILE offers significant speed advantage over LLM-as-judge methods, as extracting representations from lightweight embedding models like BERT (Devlin et al., 2019) is far faster than generating natural language outputs. Like other embedding-based metrics, this speed advantage can be further enhanced by *pre-computing* and storing representations for synthetic answers $e(\tilde{y})$ and keyword representations $e(y^{\star})$ before evaluation. By storing $e(\tilde{y})$ and $e(y^{\star})$, only the model response representations $e(y)$ and $e(N_i[y])$ need to be calculated during test time. Notably, synthetic answer generation is a *one-time* preprocessing cost per dataset; once generated, synthetic answers are reused across all model evaluations, amortizing this cost to negligible overhead when benchmarking multiple models.

## 4.2 Interpretability of SMILE scores

SMILE's semantic and lexical subscores provide practitioners with more interpretable and actionable feedback than other metrics. These subscores enable monitoring of model performance along two complementary axes: semantic content, a holistic measure of response relevance, and lexical exactness, a finer-grained measure of response quality. Importantly, SMILE allows evaluation not only at the instance-level but also at the *population-level*. Aggregating $s_s$ and $s_{\ell}$ across all test samples reveals a model's *general* strengths and weaknesses. This contrasts with LLM-as-judge methods, which offer more specific *instance-level* natural language feedback, making it hard to extract overall insights. See examples in Section C.

# 5 Experiments and results

**Benchmarks and generator models.** We assessed SMILE on established benchmarks across three domains: NLQA, VQA, and VidQA. To ensure diverse evaluation, we included three benchmarks per domain: MRQA (Fisch et al., 2019), HotpotQA (Yang et al., 2018b), and MuSiQue (Trivedi et al., 2022) for NLQA, TextVQA (Singh et al., 2019), DocVQA (Mathew et al., 2020), and POPE (Li et al., 2023b) for VQA, and TGIF (Jang et al., 2017), MSVD (Xu et al., 2017), and MSRVTT (Xu et al., 2016) for VidQA. For HotpotQA and MuSiQue, we used the standardized setup from ContextualBench (Nguyen et al., 2024). We generated responses using the following models for each domain: GPT-4o (Hurst et al., 2024) for NLQA, LLaVA-1.5 for VQA 7B (Liu et al., 2023a;b) , and Qwen2.5-VL 3B Instruct (Bai et al., 2025) for VidQA. These models were selected for their strong capabilities in producing high-quality textual responses, forming the basis of our analysis. We also evaluate on QA-Eval (Wang et al., 2023a), a large-scale *human-annotated* NLQA dataset (~10k samples) based on Natural Question (NQ) and TriviaQA (TQ), with responses from GPT-3.5 and GPT-4o. This setup enables robust comparison of SMILE against LLM judges at scale.

**Data annotation efforts.** To evaluate QA metrics, we assessed their alignment with human judgments using a *golden evaluation set*. We constructed this set by sampling model outputs from the nine benchmarks (three per domain), randomly selecting 25 input-output pairs per dataset for annotation. Four annotators (authors of the paper with native level English) evaluated the generated outputs based on a predefined rubric, considering correctness, relevance, and clarity. To ensure unbiased evaluation, all annotations were conducted in a double-blinded manner—annotators were not informed which model generated each answer nor which dataset each question originated from, and questions and outputs were shuffled randomly across all benchmarks. Given potential ambiguity, annotators used a 3 point scale: clearly incorrect, unclear, clearly correct. To check annotation quality, we calculated Krippendorff's alpha (Krippendorff, 2011), achieving a score of **0.71**, indicating substantial inter-annotator agreement. This high agreement confirms the reliability of our annotations, so we proceed with it as the basis of our evaluation.

**Baselines and metrics.** We compared SMILE with established metrics, including traditional NLP measures: ROUGE-L, METEOR and Exact and Easy Match; alongside embedding-based similarity metrics: BERTScore (with Roberta-large) and sBERT cosine similarity[1]. Following Maaz et al. (2024), we also employed GPT-4o and GPT-3.5-Turbo as judge models, prompting them for a 0-5 score and a binary yes/no prediction. For all baselines, we provide detailed implementation details in Section A, including judge model prompts.

**SMILE implementation.** We choose Llama-3.2-3B-Instruct as our synthetic answer generator $g$ and ember-V1[2] as our embedding model $e$. This combination is computationally lightweight: the 335M parameter ember-v1 can run inference on a CPU, and generating responses with the 3B Llama model requires $< 10$GB of VRAM. Furthermore, our ablation study (Section 5.4) shows that larger models offer only marginal performance improvements, highlighting SMILE's inherent lightweight nature. SMILE scores, similar to GPT-4o, are discretized into six bins (0–5), with scores $\geq 4$ considered correct. The N-gram value is dynamically set based on ground truth answer length, and the parameter $w$ fixed to 0.5.

## 5.1 Main experimental results

Using our golden evaluation set, we compare SMILE against existing baseline metrics. To holistically assess evaluators-human agreement, we computed Pearson correlation and Kendall's Tau-b from Human Accuracy. Pearson correlation and Kendall's Tau measure agreement with human annotations on the instance level, ranging from –1 (perfect disagreement) to +1 (perfect agreement). Kendall's Tau-b, focuses on ranking consistency and accounts for ties in the data. We report results in two evaluation settings to separate formatting effects from metric behavior: *Orig* uses the original ground-truth references (the common practical setup), and *Syn* uses SMILE's synthetic canonicalized references.

Pearson correlation results are presented in Table 1. SMILE achieves the best overall correlation among all evaluation metrics, consistently outperforming traditional NLG metrics (ROUGE-L, METEOR, BERTScore,

---

[1]https://huggingface.co/sentence-transformers/all-roberta-large-v1
[2]https://huggingface.co/llmrails/ember-v1

| Metric | Ref. | Video QA: Qwen2.5 | | | Visual QA: llava 1.5 7B | | | Language QA: GPT-4o | | | Overall |
|---|---|---|---|---|---|---|---|---|---|---|---|
| | | TGIF | MSVD | MSRVTT | TextVQA | DocVQA | POPE | MRQA | HotpotQA | MUSIQUE | |
| Exact Match | Orig | nan | nan | nan | nan | nan | 0.099 | nan | 0.109 | 0.147 | 0.118 |
| | Syn | nan | nan | nan | nan | nan | 0.099 | nan | 0.109 | 0.147 | 0.118 |
| Easy Match | Orig | 0.793 | 0.379 | 0.290 | 0.795 | 0.375 | 0.451 | 0.676 | 0.657 | 0.890 | 0.590 |
| | Syn | nan | nan | nan | nan | nan | 0.143 | 0.129 | 0.185 | 0.185 | 0.161 |
| ROUGE-L | Orig | 0.603 | 0.442 | 0.217 | 0.596 | 0.705 | 0.001 | 0.361 | 0.603 | 0.445 | 0.441 |
| | Syn | 0.625 | 0.372 | -0.340 | 0.081 | 0.228 | 0.518 | 0.284 | 0.042 | -0.053 | 0.283 |
| METEOR | Orig | 0.663 | 0.488 | 0.438 | 0.667 | 0.747 | 0.086 | 0.528 | 0.664 | 0.611 | 0.544 |
| | Syn | 0.617 | 0.451 | -0.311 | 0.264 | 0.314 | 0.414 | 0.283 | 0.051 | -0.046 | 0.306 |
| BERTScore | Orig | 0.421 | 0.331 | 0.110 | 0.398 | 0.677 | 0.164 | 0.310 | 0.620 | 0.411 | 0.382 |
| | Syn | 0.674 | 0.497 | -0.247 | 0.206 | 0.345 | 0.434 | 0.223 | 0.094 | 0.194 | 0.324 |
| sBERT | Orig | 0.472 | 0.592 | 0.315 | 0.602 | 0.852 | -0.164 | 0.352 | 0.664 | 0.363 | 0.486 |
| | Syn | 0.806 | 0.579 | 0.020 | 0.419 | 0.426 | 0.541 | 0.037 | 0.203 | 0.195 | 0.358 |
| BLEURT | Orig | -0.101 | 0.256 | -0.004 | 0.483 | 0.705 | -0.488 | 0.354 | 0.558 | 0.454 | 0.378 |
| | Syn | 0.692 | 0.517 | -0.093 | 0.517 | 0.504 | 0.366 | 0.174 | 0.158 | 0.136 | 0.351 |
| Moverscore | Orig | 0.613 | 0.390 | 0.278 | 0.338 | 0.671 | 0.053 | 0.242 | 0.379 | 0.231 | 0.355 |
| | Syn | 0.629 | 0.388 | -0.331 | 0.296 | 0.426 | 0.414 | 0.190 | 0.001 | 0.074 | 0.305 |
| GPT-3.5 | Orig | 0.825 | 0.609 | 0.350 | 0.663 | 0.803 | 0.422 | 0.810 | 0.668 | 0.525 | 0.631 |
| | Syn | 0.742 | **0.845** | 0.059 | 0.584 | 0.702 | 0.210 | 0.389 | 0.234 | 0.310 | 0.453 |
| GPT-4o | Orig | 0.778 | 0.687 | **0.627** | **0.859** | 0.761 | 0.699 | 0.294 | 0.678 | 0.764 | 0.683 |
| | Syn | **0.898** | 0.766 | 0.358 | 0.844 | 0.904 | **0.874** | **0.944** | 0.774 | 0.690 | 0.784 |
| **SMILE** | Orig | 0.800 | 0.564 | 0.552 | 0.784 | **0.914** | 0.469 | 0.860 | **0.952** | **0.950** | 0.761 |
| | Syn | 0.847 | 0.607 | 0.496 | 0.796 | 0.909 | 0.763 | 0.839 | 0.902 | 0.948 | **0.790** |

Table 1: **Pearson correlation with human judgments (↑) across Video, Visual, and Language QA.** For each metric we report two settings: *Orig* (evaluated against original ground-truth references) and *Syn* (evaluated against SMILE's synthetic references), shown for all datasets and Overall. SMILE achieves the strongest overall correlation across modalities. NaN values indicate zero variance in metric scores, making correlation undefined.

sBERT, BLEURT, Moverscore) and GPT-3.5 by substantial margins. While GPT-4o achieves slightly higher correlation in some multimodal settings (Video QA, Visual QA), SMILE demonstrates superior performance in Language QA and attains the highest overall correlation, indicating agreement with human evaluations.

Kendall's Tau-b results, presented in Table 2, further establish SMILE's superior ranking agreement with human judgments. SMILE achieves the highest Kendall's Tau-b in 5 out of 6 domain categories, surpassing both GPT-4o and GPT-3.5. This dominance in ranking agreement underscores SMILE's exceptional ability to order generated responses in a way that closely mirrors human annotated rankings.

Finally, we evaluate SMILE and GPT-3.5/4o as evaluators on QA-Eval, a human-annotated benchmark containing manually annotated correctness judgments for question-answer pairs. We use two prompting variants: (1) original prompt (based on Maaz et al. (2024)), and (2) extract-style prompt (asks LLM to extract short answer first). As shown in Table 3, SMILE consistently outperforms GPT-3.5 and closely matches GPT-4o. Notably, LLMs degrade under the extract prompt, highlighting SMILE's robustness and prompt independence.

**Impact of Synthetic Canonicalized References.** To analyze the impact of canonicalized references, we compare results using *Orig* and *Syn* answers. Across datasets and modalities, traditional metrics stagnate or degrade under *Syn*, whereas SMILE remains stable and typically strongest. Canonicalization therefore does not "rescue" baseline metrics; SMILE's agreement with humans is robust to the choice of reference.

| Metric | Ref. | Video QA: Qwen2.5 | | | Visual QA: llava 1.5 7B | | | Language QA: GPT-4o | | | Overall |
|---|---|---|---|---|---|---|---|---|---|---|---|
| | | TGIF | MSVD | MSRVTT | TextVQA | DocVQA | POPE | MRQA | HotpotQA | MUSIQUE | |
| Exact Match | Orig | nan | nan | nan | nan | nan | 0.100 | nan | 0.109 | 0.147 | 0.119 |
| | Syn | nan | nan | nan | nan | nan | 0.100 | nan | 0.109 | 0.147 | 0.119 |
| Easy Match | Orig | 0.765 | 0.414 | 0.295 | 0.773 | 0.361 | 0.420 | 0.676 | 0.657 | 0.890 | 0.583 |
| | Syn | nan | nan | nan | nan | nan | 0.145 | 0.129 | 0.185 | 0.185 | 0.161 |
| ROUGE-L | Orig | 0.598 | 0.509 | 0.248 | 0.614 | 0.696 | 0.162 | 0.390 | 0.496 | 0.554 | 0.474 |
| | Syn | 0.591 | 0.296 | -0.264 | 0.052 | 0.182 | 0.381 | 0.223 | 0.014 | -0.070 | 0.230 |
| METEOR | Orig | 0.592 | 0.511 | 0.446 | 0.644 | 0.686 | 0.161 | 0.396 | 0.458 | 0.582 | 0.497 |
| | Syn | 0.522 | 0.339 | -0.255 | 0.261 | 0.218 | 0.294 | 0.222 | 0.028 | -0.046 | 0.243 |
| BERTScore | Orig | 0.347 | 0.240 | 0.060 | 0.322 | 0.526 | 0.125 | 0.289 | 0.455 | 0.440 | 0.312 |
| | Syn | 0.618 | 0.411 | -0.187 | 0.131 | 0.290 | 0.408 | 0.171 | 0.043 | 0.185 | 0.272 |
| sBERT | Orig | 0.393 | 0.505 | 0.230 | 0.435 | 0.662 | -0.102 | 0.289 | 0.455 | 0.382 | 0.384 |
| | Syn | 0.674 | 0.394 | -0.043 | 0.305 | 0.336 | 0.374 | 0.051 | 0.142 | 0.231 | 0.283 |
| BLEURT | Orig | 0.019 | 0.231 | -0.034 | 0.566 | 0.526 | -0.238 | 0.323 | 0.469 | 0.520 | 0.325 |
| | Syn | 0.571 | 0.419 | -0.068 | 0.435 | 0.463 | 0.306 | 0.170 | 0.028 | 0.104 | 0.285 |
| Moverscore | Orig | 0.506 | 0.342 | 0.170 | 0.226 | 0.426 | -0.011 | 0.341 | 0.355 | 0.104 | 0.276 |
| | Syn | 0.646 | 0.300 | -0.221 | 0.226 | 0.327 | 0.328 | 0.187 | -0.057 | -0.012 | 0.256 |
| GPT-3.5 | Orig | 0.738 | 0.487 | 0.291 | 0.625 | 0.690 | 0.441 | 0.638 | 0.439 | 0.570 | 0.547 |
| | Syn | 0.668 | **0.759** | 0.033 | 0.549 | 0.642 | 0.209 | 0.341 | 0.261 | 0.300 | 0.418 |
| GPT-4o | Orig | 0.686 | 0.580 | **0.575** | **0.780** | 0.658 | **0.676** | 0.281 | 0.488 | 0.767 | 0.610 |
| | Syn | **0.797** | 0.638 | 0.272 | 0.728 | 0.754 | 0.660 | 0.614 | 0.688 | 0.589 | 0.638 |
| **SMILE** | Orig | 0.765 | 0.559 | 0.485 | 0.730 | 0.841 | 0.607 | **0.805** | **0.765** | **1** | **0.729** |
| | Syn | 0.773 | 0.595 | 0.463 | 0.652 | **0.920** | 0.607 | **0.805** | **0.765** | 0.981 | **0.729** |

Table 2: **Kendall's Tau-b (↑) ranking agreement with human judgments across Video, Visual, and Language QA.** For each metric we report two settings: *Orig* (original ground-truth references) and *Syn* (SMILE's synthetic references). Across all datasets and Overall SMILE attains the highest ranking agreement. NaN values indicate zero variance in metric scores, making correlation undefined.

| | GPT 3.5 | | GPT-4o | | Overall |
|---|---|---|---|---|---|
| | NQ | TQ | NQ | TQ | |
| GPT-3.5, original prompt | 0.756 | 0.849 | 0.713 | 0.706 | 0.756 |
| GPT-4o, original prompt | **0.865** | **0.913** | **0.815** | **0.806** | **0.850** |
| GPT-3.5, extract prompt | 0.478 | 0.572 | 0.413 | 0.440 | 0.476 |
| GPT-4o, extract prompt | 0.831 | 0.898 | 0.783 | 0.774 | 0.821 |
| **SMILE** | 0.829 | 0.889 | 0.786 | 0.760 | 0.816 |

Table 3: **Pearson Correlation with human judgment on QAEval (∼10k human-annotated samples).** SMILE shows strong agreement with human annotations, outperforming GPT-3.5 and roughly matching GPT-4o.

Short-answer examples illustrate the *Syn*-induced lexical-overlap bias: under *Syn*, traditional metrics tend to over-score verbose or even contradictory responses because increased surface overlap with the canonicalized reference can mask factual errors. On TGIF, generic non-answers receive higher scores; on POPE, fluent contradictions (e.g., "no bowl" versus "contains a bowl") are similarly over-scored. These cases show that canonicalization alone cannot fix metrics that privilege surface overlap and fluency over semantic correctness.

| | Video QA: Qwen2.5 | | | Visual QA: llava 1.5 7B | | | Language QA: GPT-4o | | | |
|---|---|---|---|---|---|---|---|---|---|---|
| | TGIF | MSVD | MSRVTT | TextVQA | DocVQA | POPE | MRQA | HotpotQA | MuSiQue | Overall |
| GPT-4o | 0.705 | 0.657 | 0.503 | 0.436 | 0.191 | 0.783 | 0.920 | 0.909 | 0.700 | **0.645** |
| Exact Match | -0.705 | -0.657 | -0.503 | -0.424 | -0.187 | -0.782 | -0.877 | -0.884 | -0.681 | 0.633 |
| Easy Match | **-0.025** | **-0.217** | **-0.106** | -0.056 | **-0.044** | **-0.021** | -0.083 | -0.152 | -0.135 | 0.093 |
| ROUGE-L | -0.705 | -0.657 | -0.503 | -0.413 | -0.179 | -0.773 | -0.648 | -0.493 | -0.553 | 0.547 |
| METEOR | -0.705 | -0.657 | -0.503 | -0.405 | -0.166 | -0.783 | -0.592 | -0.509 | -0.454 | 0.530 |
| BERTScore | 0.294 | 0.340 | 0.497 | 0.562 | 0.805 | 0.216 | **0.008** | 0.091 | 0.300 | 0.354 |
| sBERT | -0.705 | -0.657 | -0.503 | -0.379 | -0.145 | -0.780 | -0.566 | -0.399 | -0.504 | 0.515 |
| **SMILE** | -0.032 | -0.241 | 0.132 | **0.021** | 0.104 | -0.041 | **-0.008** | **-0.016** | **0.005** | **0.067** |

Table 4: **Deviation from GPT-4o accuracy** across Video, Visual, and Language QA tasks, using *complete test sets*. SMILE exhibits the smallest deviation among evaluators, closely aligning with GPT-4o.

| Metrics | Checkpoint 1 | | Checkpoint 2 | |
|---|---|---|---|---|
| | Rank | Cost($) | Rank | Cost($) |
| GPT-4o | 1 | 12 | 2 | 11.99 |
| GPT-3.5-turbo | 1 | 12 | 2 | 12.00 |
| METEOR | 1 | - | 2 | - |
| sBERT | 2 | - | 1 | - |
| SMILE | **1** | - | **2** | - |

Table 5: Checkpoint selection on TGIF video QA (Video model): Rank (1=best) and approximate evaluation cost (USD) per metric. SMILE ranks checkpoints similarly to GPT metrics, but without inference costs. "-" denotes methods without API inference cost.

SMILE mitigates this by combining sentence-level semantic similarity with keyword-level exactness and an optional exact-match component in a weighted aggregation that captures both intent and factual content. Qualitative examples in Figure 2 illustrate these patterns across TGIF, DocVQA, TextVQA, and HotpotQA

### 5.2 SMILE as a drop-in replacement for GPT-4o

Building on SMILE's alignment with human judgment, we now demonstrate its capability to supplant GPT-4o as an evaluation metric. To do so, we compare model accuracy derived from SMILE scores against that from GPT-4o-based evaluation each benchmark's *complete test-set*. We find that SMILE exhibits the lowest overall deviation among all tested methods, as summarized in Table 4. This compelling result strongly suggests SMILE is a reliable and direct alternative to resource-intensive LLM-as-judge approaches like GPT-4o.

### 5.3 Cost effective Model selection with SMILE

Selecting optimal checkpoints during ML model training is crucial for maximizing performance on downstream tasks. Traditionally, this selection process relies on not-so reliable metrics like METEOR and ROUGE, or expensive metrics such as LLM-based judge evaluations. In this experiment, we use SMILE to identify the best checkpoint. Specifically, we select two intermediate checkpoints(with similar performance) from the Video model and evaluate their performance on the TGIF benchmark. The evaluation is conducted using five metrics: GPT-4o, GPT-3.5-turbo, METEOR, sBERT and SMILE.

As per Table 5, our findings demonstrate that *SMILE selects the same optimal checkpoint (i.e. checkpoint 1) as GPT-4o and GPT-3.5-turbo.* This alignment highlights SMILE's effectiveness, emphasizing its capability to provide reliable checkpoint selection without incurring additional evaluation cost.

|  |  | Video QA | Visual QA | Language QA | Overall |
|---|---|---|---|---|---|
| *Component Ablation* | **SMILE** | **0.650** | 0.823 | 0.896 | 0.790 |
| | w/o semantic scores | 0.628 | 0.775 | 0.941 | 0.782 |
| | w/o keyword scores | 0.463 | 0.533 | 0.249 | 0.415 |
| | w/o synthetic answers | 0.639 | 0.722 | 0.921 | 0.761 |
| *Model Ablation* | **SMILE** | **0.650** | 0.823 | 0.896 | 0.790 |
| | Embedding: $GTE_{7B}$ | 0.647 | **0.824** | **0.947** | **0.806** |
| | Syn. answer: GPT-3.5-Turbo | 0.636 | 0.802 | 0.930 | 0.790 |

Table 6: **Component and model ablations.** Performance is assessed by Pearson correlation. Keyword scores are the primary contributor, highlighting the importance of lexical exactness. Embedding model scaling yields marginal ($< 2\%$) gains.

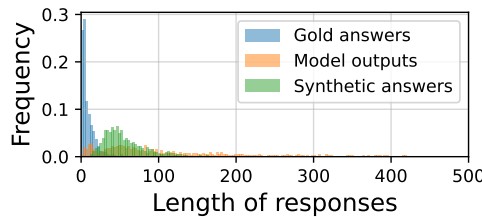

Figure 4: Length of gold answers, model outputs, and synthetic answers, across all domains and benchmarks in characters. Synthetic answers more align better with model outputs in terms of output length, enabling better semantic evaluation.

An advantage of SMILE is its substantial reduction in evaluation costs compared to GPT-based models. GPT-4o and GPT-3.5 cost's around \$12 for each checkpoint evaluation on TGIF, and the cost increases as more checkpoints and evaluation benchmarks are added. In contrast, SMILE has almost no extra cost. Therefore, adopting SMILE not only maintains performance accuracy but also significantly lowers monetary overhead, making it a highly efficient and scalable solution for checkpoint selection.

### 5.4 Ablations

This section presents an ablation study of SMILE centered on three key perspectives: (1) *Component analysis*, systematically removing steps (synthetic answer generation, semantic similarity score, keyword score) to demonstrate their individual importance, (2) *Model scaling*, examining the impact of using larger models for both synthetic answer generation and embedding, (3) *Hyperparameter tuning*, analyzing the effect of the weight $w$ in SMILE. Results are detailed in Table 6 and Figure 5.

**Component analysis.** SMILE comprises three key components: (1) semantic similarity, (2) lexical exactness, and (3) distribution alignment via a lightweight language model. Table 6 (top) summarizes the contribution of these components to SMILE's robust performance. Experiments demonstrate that *both keyword and semantic scores are essential*. Removing keyword scores significantly reduces Pearson correlation, underscoring the critical role of lexical exactness in QA evaluation. Conversely, relying solely on keyword scores neglects global structure, degrading performance notably in VidQA and VQA. Synthetic answers provide *domain-dependent* benefits: Visual QA sees a $+14\%$ improvement (gold answers are often single words while model outputs are verbose), while Language QA sees minimal change since predictions are already concise. Figure 4 illustrates this effect, showing how synthetic answers bridge the length mismatch between short gold answers and longer model outputs. Combining semantic scores, keyword scores, and synthetic answers yields robust and accurate evaluation across domains.

**Model scaling.** A key advantage of SMILE is that it offers the ability to *efficiently* run evaluation. Our model choices in Section 5 demonstrate this: SMILE at inference time requires only a 355M parameter em-

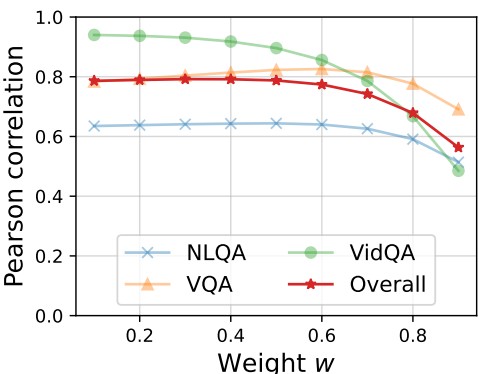

Figure 5: Sweep of $w$, which trades off lexical exactness for semantic similarity as $w$ increases. SMILE exhibits relatively stable aggregate performance for $w \leq 0.5$.

bedding model, and pre-generating synthetic answers requires only a 3B generative model. Table 6 (bottom) further establishes SMILE's lightweight nature: increasing model capacity yields minimal performance gains, if at all. Our model ablation focused on the synthetic answer generation model and the embedding model. Using GPT-3.5-Turbo instead of Llama-3.2-3B-Instruct for synthetic answers yielded comparable correlation with human judgments, indicating that *effective synthetic answer generation is achievable with smaller lightweight models*. Replacing ember-v1 with the substantially larger GTE-7B (Li et al., 2023c) embedding model resulted in only a marginal performance gain of less than 2%, despite a 20× increase in model size. This indicates that *SMILE remains effective even with lightweight embedding models*.

**Hyperparameter tuning.** SMILE's hyperparameter $w$ allows practitioners to precisely decide the impact of the semantic and keyword subscores. Specifically, as $w$ increases, more importance is given to the semantic subscore. As we show in Figure 5, overall performance is relatively stable for $w \leq 0.5$ before smoothly decreasing. This aligns with results from our component ablation study in Table 6: The keyword subscores alone exhibited relatively strong performance, while the semantic subscore fared worse. However, SMILE hyperparameter choice is relatively forgiving, with any choice of $w$ that slightly upweights the keyword subscore likely to perform well. For practitioners selecting $w$: use $w$=0.0–0.3 for factual/entity QA or verbose models (keyword matching critical); $w$=0.5–0.6 for OCR/document QA or binary yes/no questions; $w$=0.7–0.8 for visual QA with context-dependent answers; and $w$=0.5 as a robust general-purpose default.

# 6    Conclusion

We introduce SMILE, a lightweight QA evaluation metric that integrates semantic and lexical analysis to address the high cost, biases, and inconsistencies of LLM-based evaluators. Benchmarking across text, image, and video QA demonstrates SMILE's strong correlation with human judgment, surpassing traditional metrics and LLM judges like GPT-4o, while ablation studies validate the importance of its components. SMILE provides an efficient, interpretable, and robust alternative to costly LLM evaluations, effectively balancing lexical precision with semantic relevance across modalities.

## Limitations

Although SMILE offers a lightweight, interpretable, and scalable alternative to LLM-based evaluators, it has limitations. (1) SMILE is source-free and does not access context, which may cause it to miss context-dependent errors. (2) The metric relies on synthetic answers to align ground-truths with model outputs; their quality can affect scoring, especially for long-form or open-ended responses. (3) Our evaluation is limited to factoid QA; effectiveness on complex reasoning, multi-hop, or conversational QA remains unexplored. (4) SMILE's weighting parameter for balancing lexical and semantic components may require tuning for specific tasks or domains.

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

# A    Additional SMILE and baseline implementation details

## A.1    Prompt templates

As described in Section 4, we prompt the synthetic answer generator with the original question and ground-truth answer and task it with generating a synthetic answer. We provide the full prompt below.

```
### Synthetic answer generation prompts
## System prompt:
You are an intelligent chatbot designed for generating answer as a sentence from question-answer pairs.
Your task is to generate a single sentence answer using the question and the answer already provided. Here's
how you can accomplish the task:
------
##INSTRUCTIONS:
- Look at the provided answer.
- Generate a short single sentence response using the question and the answer.
- Response SHOULD ALWAYS USE THE WORDS FROM ANSWER provided.
- DO NOT USE THE QUESTION AS IT IS IN THE RESPONSE.
- Return only the response and nothing else.

## User prompt
Please phrase a short single sentence answer using question-answer pair only:
Question: {<question>}
Answer: {<answer>}
DO NOT PROVIDE ANY OTHER OUTPUT APART FROM A SINGLE SHORT SENTENCE.
```

To prompt GPT-4o and GPT-3.5 as judge models, we utilize prompts adopted from Maaz et al. (2024), as described in Section 5. We provide full prompts below.

```
### Original prompt: GPT-4o/GPT-3.5-Turbo judge prompts
## System prompts
You are an intelligent chatbot designed for evaluating the correctness of generative outputs for
question-answer pairs.
Your task is to compare the predicted answer with the correct answer and determine if they match meaningfully.
Here's how you can accomplish the task:
------
##INSTRUCTIONS:
- Focus on the meaningful match between the predicted answer and the correct answer.
- Consider synonyms or paraphrases as valid matches.
- Evaluate the correctness of the prediction compared to the answer.

## User prompt
Please evaluate the following video-based question-answer pair:
Question: {<question>}
Correct Answer: {<answer>}
Predicted Answer: {<model_output>}
Provide your evaluation only as a yes/no and score where the score is an integer value between 0 and 5, with 5
indicating the highest meaningful match.
Please generate the response in the form of a Python dictionary string with keys 'pred' and 'score', where
value of 'pred' is  a string of 'yes' or 'no' and value of 'score' is in INTEGER, not STRING.
DO NOT PROVIDE ANY OTHER OUTPUT TEXT OR EXPLANATION. Only provide the Python dictionary string.
For example, your response should look like this: {{'pred': 'yes', 'score': 4}}.
```

```
### Extract prompt: GPT-4o/GPT-3.5-Turbo judge prompts
## System prompts
You are an expert evaluator for video-based question answering systems. Your task is to judge the factual
accuracy of a predicted answer by comparing it to a correct answer. You will follow a structured evaluation
approach to ensure consistency:
------
## INSTRUCTIONS:
Step 1: Extract the key facts from the Correct Answer.
Step 2: In case the Correct Answer is a list, choose the best answer that matches the Predicted Answer.
Step 3: Extract the key facts from the Predicted Answer.
Step 4: Compare the two sets of facts and determine how consistent they are.
 - Consider paraphrasing, synonyms, and partial overlaps.
 - Ignore grammatical errors.
 - Penalize hallucinated or contradicted information.
Step 4: Based on the comparison, assign a factual accuracy score between 0 and 5 (INTEGER only), where:
```

```
 5 = Fully accurate and aligned
 4 = Mostly accurate, minor omissions or paraphrasing
 3 = Partially correct but with notable missing or incorrect info
 2 = Limited accuracy, mostly incorrect or unrelated
 1 = Completely inaccurate
 0 = No relation or total hallucination
Respond strictly in the following format:
{'score': X, 'pred':Y} where X is an integer between 0 and 5 and Y is a either 'yes'(X>3) or 'no'(X<=3). Do not
include any explanation or extra text.}

## User prompt
Evaluate the following video-based QA pair:
Question: {<question>}
Correct Answer: {<answer>}
Predicted Answer: {<model_output>}
Return your evaluation following the instructions above.
DO NOT PROVIDE ANY OTHER OUTPUT TEXT OR EXPLANATION. Only provide the Python dictionary string.
```

## A.2  SMILE text processing

As a part of text pre-processing, we perform standard text normalization on words present in ground truth answers and predictions. We first convert each string to lower case and remove all punctuation. Then, each word is lemmatized using POS-aware lemmatization to capture accurate base forms. If the resulting processed word is empty after these steps, the original lower-case word is retained.

## A.3  Metrics conversion to accuracy

For all evaluated baselines and metrics, we must convert from scores to binary correct or incorrect accuracy labels. ROUGE, METEOR, BERTScore, sBERT, and SMILE all output continuous-valued scores between 0 and 1. We apply a threshold of 0.67, considering anything above the threshold to be correct and anything below to be incorrect. The choice of 0.67 is the same as considering anything with a score of 4 or above to be correct after converting the continuous [0,1] score to a 0-5 scale with uniform binning. For GPT-3.5-Turbo and GPT-4o, the model is prompted to output a yes/no label for correctness, which we use directly.

# B  Data annotation details

## B.1  Annotation instructions

Annotators were given with detailed instructions on how to annotate responses. We adopted a 3 point scale: `clearly incorrect`, `unclear`, and `clearly correct`. We defined each of these categories as follows:

**Clearly incorrect:** The model definitively produces a response that is incorrect.

**Unclear:** The model response cannot be confirmed correct from the ground-truth answer.

**Clearly correct:** The model response can be explicitly verified as correct using the ground-truth answer.

We also defined edge case behavior:

**Extraneous information:** If the model response correctly answers the question, but includes other information that may or may not be factual, we consider the response `clearly correct`. As a concrete example, for question "What brand of soda is in this picture" with ground-truth "Coca-Cola", we consider the model response "Coca-Cola is in this picture. It is the most popular soda in the world by unit sales and has over 60 different flavors" to be correct, even though it contains extraneous factually verifiable information.

**Synonyms or ambiguous subjects:** We consider a model response that answers the question using an ambiguous subject to be `unclear`. As a concrete example, for question "who describes a video game??" with ground-truth "man", we consider the model response "person" to be unclear, as it does not describe in sufficient detail the person.

### B.2 Annotation aggregation and conversion to accuracy labels

We collected responses from four annotators. To aggregate individual annotations into a single label, we utilized majority vote, employing random tie-breaking as needed. To form final accuracy labels, we consider `clearly correct` responses to be accurate and consider `unclear` and `clearly incorrect` responses to be inaccurate.

## C SMILE Interpretability Examples

As discussed in Section 4.2, we provide detailed SMILE subscores in Figure 2. In the TGIF example from Figure 2, the model output shows a high semantic score $s_s$ (Equation (1)), reflecting strong relevance to the synthetic answer. The lexical relevance score $s_l$ (Equation (2)) is also high, indicating a perfect overlap with the ground truth. To clarify which word contributes most to the keyword score, we also return the word(s) with the maximum similarity ("max sim words"). These components together offer actionable insights into model strengths and weaknesses, helping guide targeted improvements.

## D Supplement ablation results

In this section, we present additional plots to supplement our Model Ablation described in Section 5.4. Specifically, we include scatter plots and distribution plots to further illustrate the performance difference when varying model choices for synthetic answer generation and embedding.

### D.1 Synthetic answer generation ablation

Referring to Figure 6, we see a very strong linear correlation between the two sets of generated synthetic answers and thus backs our claim that *generating synthetic answers is a fairly simple task* as mentioned in Section 5.4. Figure 7, further bolster our claim, and highlights that the 'avg score' distribution remains very similar, hence we see a marginal difference in the performance as reported in table 6.

### D.2 Embedding model ablation

Figure 8 and Figure 9 provides insight into the performance variation observed in Table 6, highlighting that keyword scores exhibit greater sensitivity to the choice of embedding model compared to sentence scores.

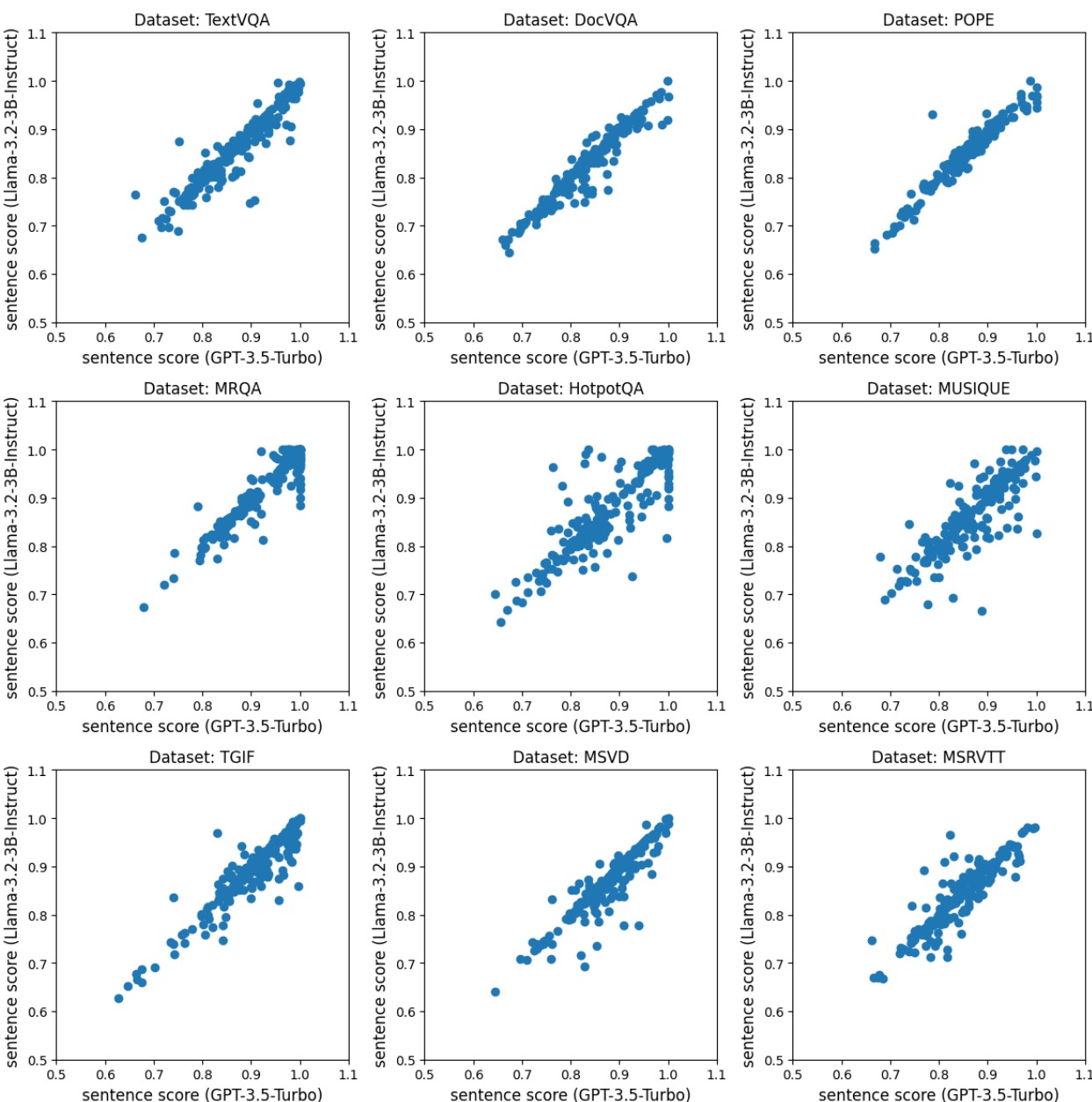

Figure 6: Distribution analysis of SMILE sentence embedding scores across different synthetic answer sets. A strong linear relationship is observed between the two synthetic answer sets, indicating that synthetic answers can reliably be generated using any state-of-the-art generation model.

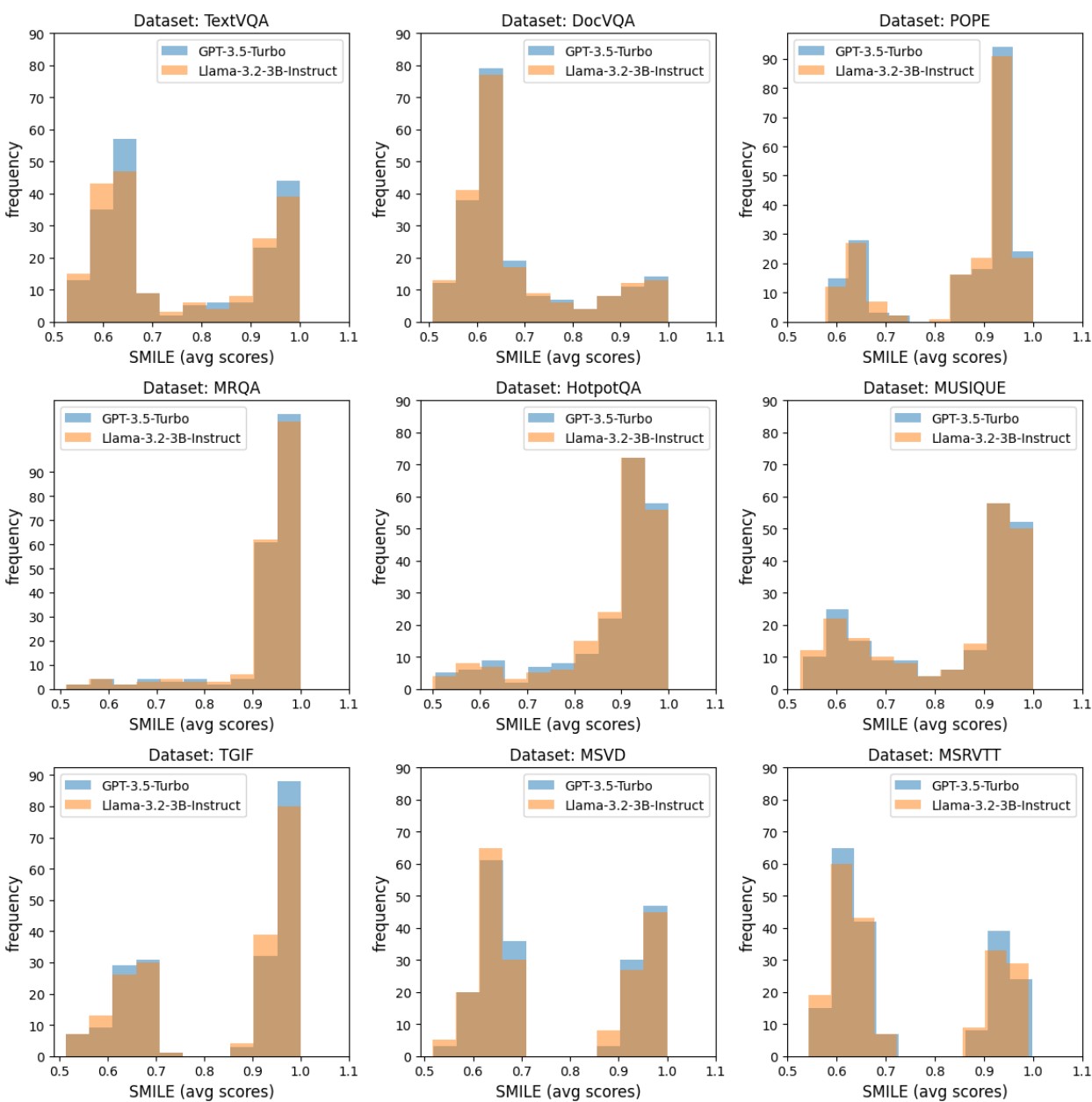

Figure 7: Distribution analysis of 'SMILE avg scores' across different synthetic answer sets. We see a very similar score distribution, highlighting the fact the performance remains very similar.

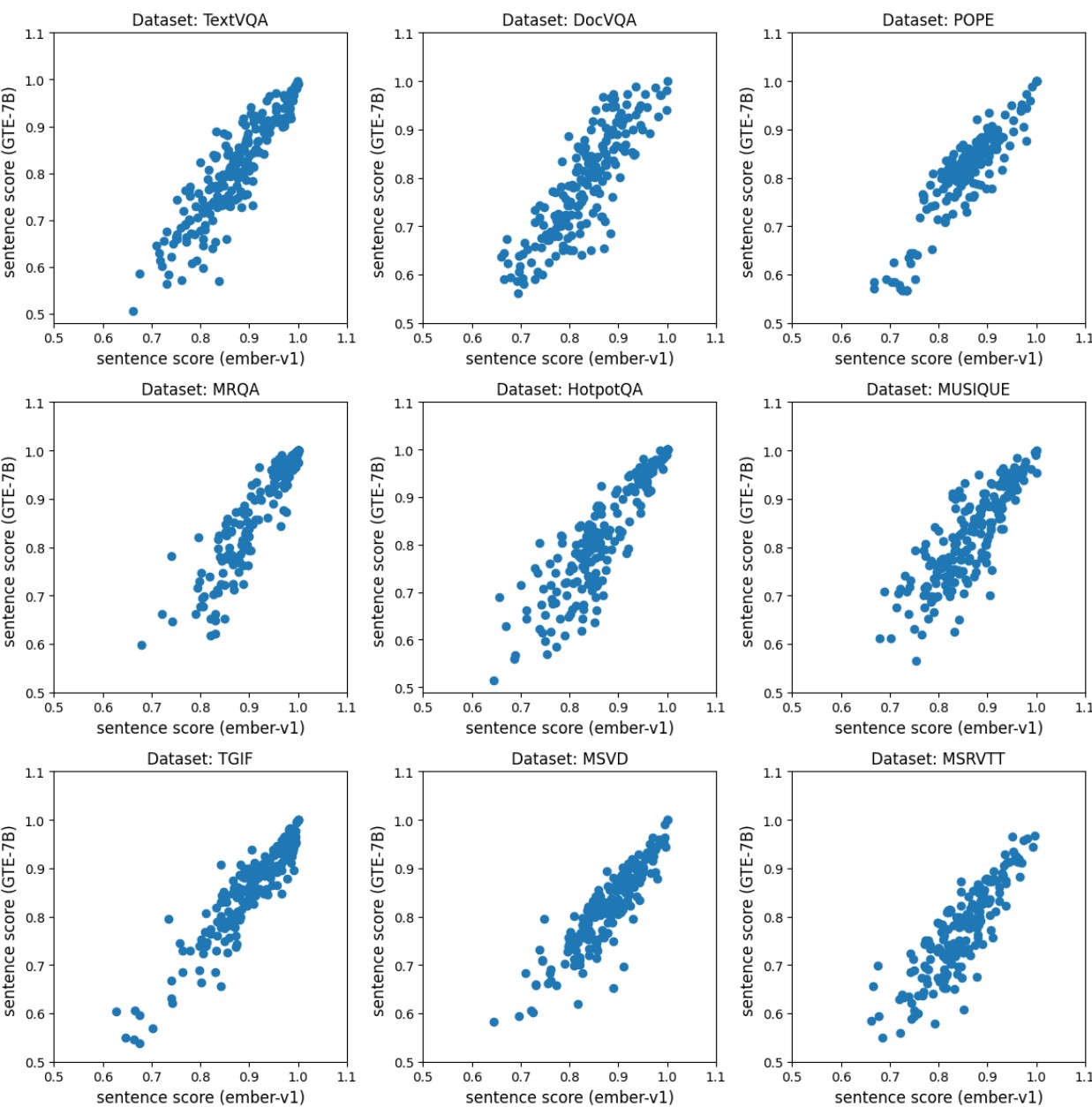

Figure 8: Analyzing Sentence score distributions using different embedding models. Sentence scores show stronger linear correlation, indicating that it is robust to change in embedding model.

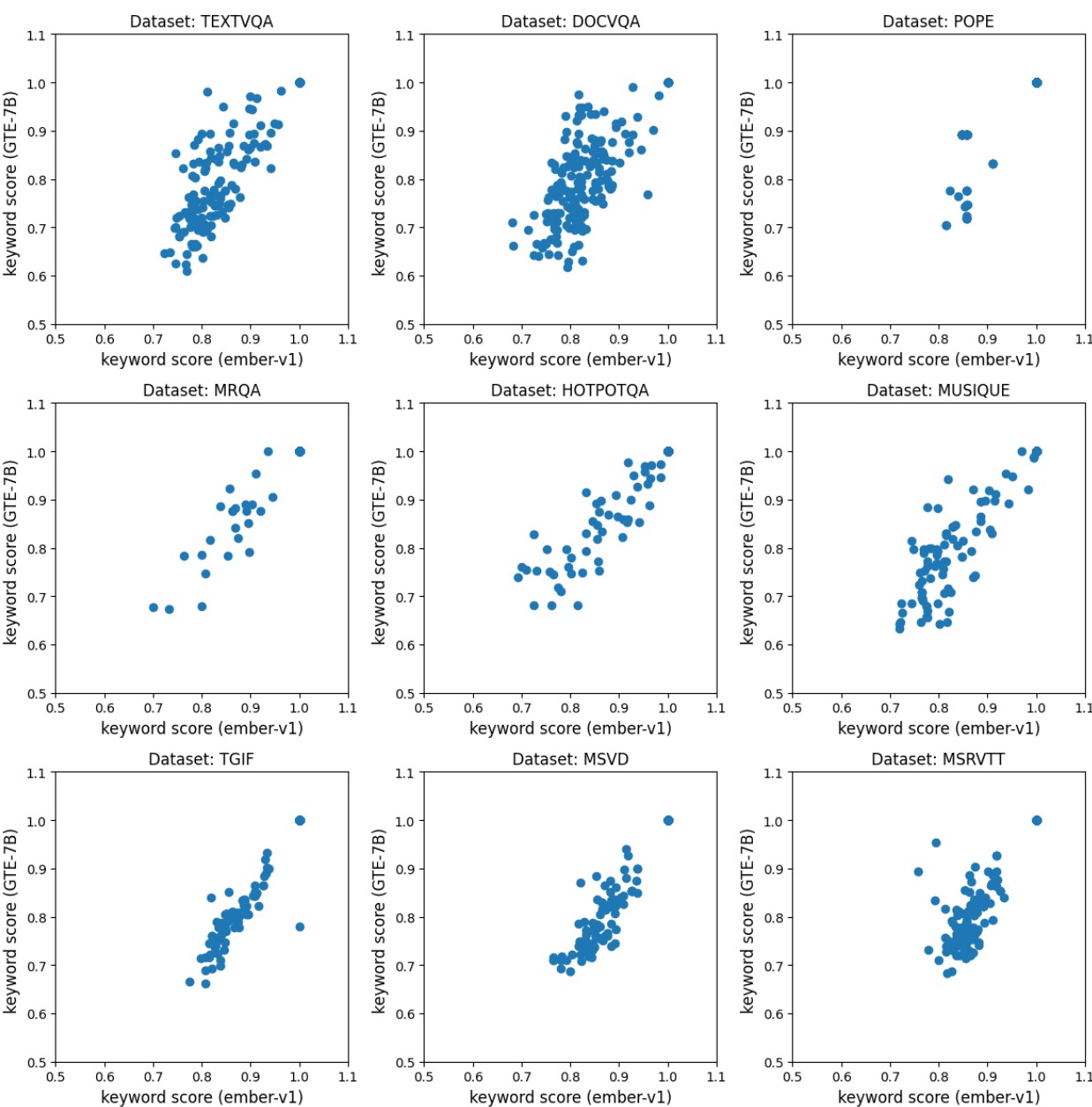

Figure 9: Analyzing Keyword score distributions using different embedding models. Keyword scores show a linear correlation, but has some added noise, indicating that it is more sensitive to changes in embedding models.

