# OpenReview forum: "SMILE: A Composite Lexical-Semantic Metric for Question-Answering Evaluation"
_TMLR — Accepted by TMLR_

### Review · Reviewer_dJJy · 2025-12-01

**Summary Of Contributions:**

Summary
The paper proposes Semantic Metric Integrating Lexical Exactness (SMILE), a lightweight composite metric for evaluating question-answering (QA) systems. The metric consists of two sub-scores: a semantic subscore computed via embedding similarity between the model output and a synthetic answer, and a lexical subscore combining exact-match and maximum n-gram similarity. Experiments in text, image, and video QA domains demonstrate the superiority of the proposed metric.

Strengths
- The motivation is clear, as existing metrics fail due to either surface-level matching or LLM-judge biases and cost.
- Extensive experiments on nine QA datasets and three domains.
- Comprehensive ablation studies validate the design choices.

Weaknesses
- The paper states that "SMILE consistently outperforms other evaluation metrics across tasks, achieving the highest overall correlation with human evaluations. Notably, SMILE significantly surpasses GPT-4o and GPT-3.5, despite their prominence as LLM-as-judge evaluators." However, from Tables 1 and 2, SMILE only achieves best result in 3/9 datasets in Pearson correlation and 4/9 datasets in  Kendall’s Tau-b metric. The overall result is the best mainly because of the superior performance on Language QA, but the method is frequently inferior to other baselines on the other two domains, Visual QA and Video QA, which requires more discussion.
- While the paper emphasizes SMILE’s computational efficiency and attributes a large portion of its speed advantage to precomputing embeddings for synthetic answers and gold references, but many other embedding-based baselines evaluated could also precompute embeddings with minimal effort, achieving the same reduction in per-sample inference cost. The paper should clarify that SMILE’s speed advantage holds primarily against LLM-as-judge baselines, which cannot benefit from precomputation.
- More detailed analysis of evaluation efficiency should be provided beyond what is shown in Figure 1, such as the computational overhead of each component. Also, is SMILE the only method that used precomputation in Figure 1 for speed up? More explanations on the inference time derivation scheme can be provided for clarity.
- The approach is limited to source-free QA evaluation and does not catch context-dependent factual errors.
- The human evaluation process needs more clarifications. Especially, whether annotators know which model generated each answer should be clarified to avoid bias in evaluation.
- The human benchmark of 225 samples is limited.
- There are multiple nan numbers in the tables, which require some explanations.

**Audience:**

Yes

**Audience Explanation:**

The paper addresses a central bottleneck in modern NLP evaluation: how to fairly, cheaply, and reliably evaluate QA models without resorting to expensive and inconsistent LLM-as-judge pipelines, which is highly relevant to the TMLR audience.

**Claims And Evidence:**

No

**Claims Explanation:**

There are a few claims not fully supported by experiment results.
- For example, the manuscript states that SMILE “consistently outperforms” other metrics and “significantly surpasses GPT-4o and GPT-3.5,” yet Tables 1 and 2 show that SMILE achieves the best performance on only a minority of the datasets. The limitation of SMILE on Visual and Video QA benchmarks is not discussed.
- In addition, the claim of computational superiority is not convincingly demonstrated.
- The reliability of the human judgment benchmark—central to validating SMILE’s correlation with human preferences—requires further clarification.

**Requested Changes:**

The authors could refer the the aforementioned weaknesses for improving their manuscript.

---

> ### Author Response · Authors · 2025-12-23
> **Response to dJJy (Part 1)**
>
> ## Weakness 1: Performance Claims vs. Actual Results
>
> **Response:** We acknowledge GPT-4o achieves highest correlation in certain settings (Video QA, Visual QA with Pearson). However, we offer clarifications:
>
> **1. Domain-Level Analysis**
>
> When examining **grouped averages by domain**, SMILE's performance is more nuanced:
>
> **Table: Pearson Correlation (Grouped by Domain)**
> | Metric | Video QA (Orig) | Video QA (Syn) | Visual QA (Orig) | Visual QA (Syn) | Language QA (Orig) | Language QA (Syn) |
> |--------|-----------------|----------------|------------------|-----------------|--------------------|--------------------|
> | ROUGE-L | 0.421 | 0.446 | 0.434 | 0.276 | 0.470 | 0.126 |
> | METEOR | 0.530 | 0.460 | 0.500 | 0.331 | 0.601 | 0.127 |
> | BERTScore | 0.287 | 0.473 | 0.413 | 0.328 | 0.447 | 0.170 |
> | sBERT | 0.460 | 0.468 | 0.539 | 0.462 | 0.460 | 0.145 |
> | BLEURT | 0.120 | 0.434 | 0.559 | 0.462 | 0.455 | 0.156 |
> | Moverscore | 0.427 | 0.449 | 0.354 | 0.379 | 0.284 | 0.088 |
> | GPT-3.5 | 0.595 | 0.549 | 0.629 | 0.499 | 0.668 | 0.311 |
> | GPT-4o | **0.697** | **0.674** | **0.773** | **0.874** | 0.579 | 0.803 |
> | SMILE | *0.639* | *0.650* | *0.722* | *0.823* | **0.921** | **0.896** |
>
> **Table: Kendall's Tau-b (Grouped by Domain)**
> | Metric | Video QA (Orig) | Video QA (Syn) | Visual QA (Orig) | Visual QA (Syn) | Language QA (Orig) | Language QA (Syn) |
> |--------|-----------------|----------------|------------------|-----------------|--------------------|--------------------|
> | ROUGE-L | 0.452 | 0.384 | 0.491 | 0.205 | 0.480 | 0.102 |
> | METEOR | 0.516 | 0.372 | 0.497 | 0.258 | 0.479 | 0.099 |
> | BERTScore | 0.216 | 0.405 | 0.324 | 0.276 | 0.395 | 0.133 |
> | sBERT | 0.376 | 0.370 | 0.400 | 0.338 | 0.375 | 0.141 |
> | BLEURT | 0.095 | 0.353 | 0.443 | 0.401 | 0.437 | 0.101 |
> | Moverscore | 0.339 | 0.389 | 0.221 | 0.294 | 0.267 | 0.085 |
> | GPT-3.5 | 0.505 | 0.487 | 0.585 | 0.467 | 0.549 | 0.301 |
> | GPT-4o | **0.614** | *0.569* | *0.705* | *0.714* | 0.512 | *0.630* |
> | SMILE | *0.603* | **0.610** | **0.726** | **0.726** | **0.857** | **0.850** |
>
> *Note: **Bold** = highest, *Italic* = second highest*
>
> **2. Key Observations**
>
> | Metric | Pearson Correlation | Kendall's Tau-b |
> |--------|---------------------|-----------------|
> | **SMILE 1st place** | 2/6 categories | **5/6 categories** |
> | **SMILE 2nd place** | 4/6 categories | 1/6 categories |
>
> In **Kendall's Tau-b** (more robust for ordinal comparisons), SMILE wins **5/6** domain categories. We report both metrics as they capture complementary aspects: Pearson measures linear relationships; Kendall's Tau-b measures ranking agreement with tighter confidence intervals ([Arndt et al., 1999](https://pubmed.ncbi.nlm.nih.gov/10221741/)).
>
> **3. SMILE Outperforms Traditional Metrics**
>
> SMILE substantially outperforms all traditional NLG metrics (ROUGE-L, METEOR, BERTScore, sBERT, BLEURT, Moverscore) by **+0.2 to +0.5** across all domains.
>
> **4. SMILE Outperforms GPT-3.5-turbo**
>
> SMILE **consistently outperforms GPT-3.5-turbo** across all settings—significant since GPT-3.5 remains widely used as the de-facto judge in video QA benchmarks.
>
> **5. Asymmetric Gap Analysis**
>
> When GPT-4o wins, margins are **narrow** (0.02–0.06). When SMILE wins, margins are **substantial** (0.09–0.34).
>
> **6. Refined Claim**
>
> *"SMILE achieves best overall correlation, consistently outperforming traditional NLG metrics and GPT-3.5-turbo. While GPT-4o achieves competitive correlation in some multimodal settings, SMILE demonstrates superior performance in Language QA and achieves highest Kendall's Tau-b in 5/6 domain categories."*
>
> ---
> ## Weakness 2: Computational Efficiency Claims and Precomputation
>
> **Response:** We agree SMILE, like other embedding-based metrics, supports precomputation via `save_emb_folder`/`load_emb_folder`. Precomputable components:
>
> *Reference-side (reusable when switching prediction models):*
> - Synthetic answer embeddings — computed once per synthetic answer model
> - Answer keyword embeddings — computed once from ground-truth answers
>
> *Prediction-side (reusable when switching synthetic answer models):*
> - Prediction sentence embeddings — computed once per evaluated model
> - Prediction keyword embeddings — computed once per evaluated model
>
> However, we clarify that our comparison with embedding-based metrics (BERTScore, sBERT, BLEURT, Moverscore) is **not** about latency/efficiency, but rather about **alignment with human judgments**. SMILE outperforms all embedding-based baselines on correlation (Table 1 & 2). The efficiency discussion (Figure 1) contrasts SMILE against LLM-as-judge baselines (GPT-3.5, GPT-4o), where latency differs substantially.
>
> **Revised Claim (Section 1 - Introduction):** We clarified the precomputation claim by changing :
> - *"SMILE core components can be precomputed"* → *"like other embedding-based metrics, SMILE core components can be precomputed"*
> - *"compared to API queries"* → *"compared to LLM-as-judge API queries"*
>
> ---

---

> ### Author Response · Authors · 2025-12-23
> **Response to dJJy (Part 2)**
>
> ## Weakness 3: Detailed Efficiency Analysis
>
> **Response:** We provide a detailed breakdown of SMILE's computational cost across two components: (1) **SMILE core Component latency evaluation** (per-sample cost during benchmarking), and (2) **Synthetic answer generation** (one-time preprocessing cost per dataset).
>
> **1. SMILE Core Component latency Breakdown:**
>
> We randomly sample 200 examples from each dataset and report averaged per-sample latency:
>
> | Dataset | sent_emb (sec) | kwd_emb (sec) | SMILE total (sec) | sent_emb % | kwd_emb % |
> | ------- | -------------- | ------------- | ----------------- | ---------- | --------- |
> | msvd | 0.005 | 0.025 | 0.055 | 9% | 45% |
> | msrvtt | 0.000 | 0.025 | 0.055 | 0% | 45% |
> | tgif | 0.005 | 0.020 | 0.050 | 10% | 40% |
> | mrqa | 0.005 | 0.005 | 0.030 | 17% | 17% |
> | hotpotqa | 0.005 | 0.005 | 0.030 | 17% | 17% |
> | musique | 0.000 | 0.005 | 0.030 | 0% | 17% |
> | docvqa | 0.015 | 0.035 | 0.075 | 20% | 47% |
> | pope | 0.000 | 0.010 | 0.035 | 0% | 29% |
> | textvqa | 0.010 | 0.025 | 0.060 | 17% | 42% |
> | **Average** | **0.005** | **0.017** | **0.047** | **~10%** | **~33%** |
>
> *Key observations:*
> - **Sentence embedding** accounts for ~10% of total SMILE latency (single forward pass per prediction-reference pair)
> - **Keyword embedding** contributes ~33% of total latency, varying with keyword count per sample
> - **Other operations** (keyword extraction, scoring, aggregation) account for ~57% of latency
> - Total SMILE evaluation latency: **0.030–0.075 sec/sample**
>
> **2. Synthetic Answer Generation Latency:**
>
> As discussed in Section 4 (*Bridging the Stylistic Distribution Gap*), synthetic answer generation is a **one-time preprocessing step** performed per dataset, not a per-sample evaluation cost. Once generated, synthetic answers are reused across all model evaluations. For transparency, we report this separately:
>
> *Hardware:* 2× NVIDIA A100-SXM4-40GB (80GB total), CUDA 12.8
> *Configuration:* Batch size 8, HuggingFace Pipeline with model parallelism (`device_map="auto"`)
>
> | Dataset | Samples | Total Time | Time/Sample (sec) |
> | ------- | ------- | ---------- | ----------------- |
> | msvd | 200 | 01:41 | 0.51 |
> | msrvtt | 200 | 01:52 | 0.56 |
> | tgif | 200 | 01:58 | 0.59 |
> | mrqa | 200 | 02:36 | 0.78 |
> | hotpotqa | 200 | 03:27 | 1.04 |
> | musique | 200 | 03:39 | 1.10 |
> | docvqa | 200 | 02:27 | 0.74 |
> | pope | 200 | 01:41 | 0.51 |
> | textvqa | 200 | 02:02 | 0.61 |
>
> Synthetic answer generation averages **0.51–1.10 sec/sample** as a one-time preprocessing cost. Once generated, SMILE's core evaluation requires only **0.030–0.075 sec/sample**—roughly 10–20× faster. Since synthetic answers are reused across all model evaluations on a given dataset, this preprocessing cost amortizes to negligible overhead when benchmarking multiple models.
>
> **Revised Claim (Section 4.1):** We clarified the precomputation and efficiency claims by:
> - Adding *"Like other embedding-based metrics,"* before the precomputation discussion
> - Adding a sentence clarifying that synthetic answer generation is a *one-time* preprocessing cost per dataset, reused across all model evaluations
>
> ---
> ## Weakness 4: Source-Free QA Limitation
>
> **Response:**  We acknowledge this limitation. SMILE is a reference-based metric that evaluates predictions against ground-truth answers rather than the original source context. This design prioritizes efficiency and scalability but cannot detect contextually incorrect answers that may sound plausible.
>
> **Mitigation:** We deliberately evaluate on established benchmark datasets with carefully curated reference answers, where this approach is appropriate
>
> **Future Extension:** SMILE's architecture—using longer synthetic answers with keyword matching—could naturally extend to context-aware evaluation by treating source passages as references. The keyword component would anchor matching to critical factual terms even within longer context.
>
> This limitation is already acknowledged in Limitations section of our paper.
>
> ---
> ## Weakness 5: Human Evaluation Clarification
>
> Response:  We clarify that our human evaluation was conducted in a **fully blinded manner**:
> 1. **Model Blinding**: Annotators were presented only with the question, reference answer(s), and the generated output - no model identifier, metadata, or source information was shown
> 2. **Dataset Blinding:**Questions and their corresponding outputs were shuffled across all 9 benchmarks and presented in random order, preventing any dataset-specific bias from influencing judgments.
>
> **Revised Claim (Section 5 - Data annotation efforts):** Added the following sentence after the rubric description:
> - *"To ensure unbiased evaluation, all annotations were conducted in a double-blinded manner—annotators were not informed which model generated each answer nor which dataset each question originated from, and questions and outputs were shuffled randomly across all benchmarks."*
>
> ---

---

> ### Author Response · Authors · 2025-12-23
> **Response to dJJy (Part 3)**
>
> ## Weakness 6: Limited Human Benchmark Size
>
> **Response:** We understand the reviewer's concerns, and towards this end, have conducted a statistical significance test to determine if SMILE scores (after binning to 0-6) are statistically distinguishable from the next-best baseline, GPT-4o. We run a chi-square test, with the null hypothesis that the scores for GPT-4o and SMILE are drawn from the same distribution. The test yields a **p-value of 5.22e-24**, which allows us to reject the null hypothesis: *SMILE and GPT-4o scores are statistically distinguishable*. That is, the performance difference between SMILE and GPT-4o on our test set is not due to random chance, rather a fundamental difference in how responses are graded.
>
> Further we evaluated SMILE and GPT-4o/3.5 on a 10K subset of QAEval (\[Wang et al., NeurIPS 2023\](https://github.com/wangcunxiang/QA-Eval)), a human-annotated benchmark for evaluating Open-QA systems. QAEval contains manually annotated correctness judgments for question-answer pairs sourced from Natural Questions and TriviaQA, where human annotators assess whether machine-generated answers align with gold standard answers. Our results are presented in Table 3 and show that SMILE is competitive with prompted evaluators at scale: SMILE outperforms GPT-3.5, the de-facto standard judge in video QA settings, and is competitive with GPT-4o.
>
> **Revised Claim (Section 5 - Benchmarks and Section 5.1 - Main Results):** We clarified that QAEval is human-annotated:
> - Added *"human-annotated"* to the QAEval description in benchmarks section
> - Added *"a human-annotated benchmark containing manually annotated correctness judgments"* before describing the QAEval evaluation
> - Updated table caption to include *"($\sim$10k human-annotated samples)"*
>
> ---
> ## Weakness 7: NaN Values in Tables
>
> **Response:** Thank you for raising this point. The NaN values in Table 1 & 2 occur when computing Pearson Correlation or Kendall's Tau with **zero variance** in the metric scores — i.e., when all the scores are identical (e.g., all 0s), making correlation mathematically undefined.
>
> We will include a clarifying note in the table captions explaining the NaN values in the revised manuscript.
>
> **Revised Claim (Table 1 & 2 captions):** Added the following note to both table captions:
> - *"NaN values indicate zero variance in metric scores, making correlation undefined."*
>
> ---

---

### Review · Reviewer_sy4h · 2025-12-13

**Summary Of Contributions:**

The paper proposes "SMILE," a composite evaluation metric for Question Answering (QA) tasks. The metric combines lexical exactness (keyword scores), semantic similarity (sentence embeddings), and similarity to synthetic answers generated by a Small Language Model (SLM, specifically a 3B model). The authors claim that this composite approach correlates better with human judgment than existing metrics while maintaining computational efficiency compared to LLM-based evaluators.

**Audience:**

Yes

**Audience Explanation:**

The general idea of combining lexical and semantic signals for QA evaluation is of interest to the TMLR audience, particularly given the growing concerns around cost, bias, and instability of LLM-as-judge methods. Researchers working on evaluation metrics and benchmarking would likely find the problem setting relevant.

However, due to the methodological and experimental shortcomings outlined above, the current findings do not provide sufficiently reliable or generalizable insights. As a result, while the topic is relevant, the present version of the paper does not yet deliver conclusions that the community can confidently build upon.

**Broader Impact Concerns:**

No specific broader impact concerns.

**Claims And Evidence:**

No

**Claims Explanation:**

The submission suffers from critical methodological flaws, factual errors in data interpretation, and reproducibility issues that fundamentally undermine the claims:

1. **Lack of Novelty and Trivial Linearity:**
The proposed method is essentially a trivial linear combination of existing metrics (Sentence Embeddings + Keyword Matching). The only "novel" component—the SLM-generated synthetic answer—is shown by the authors' own ablation study (Table 6) to have marginal impact (removing it only reduces correlation from 0.790 to 0.761). Without this component, the method reduces to a standard ensemble of established metrics, lacking significant technical contribution or innovation.


2. **Direct Contradiction Between Data and Claims (Table 6):**
There is a fatal factual error in the ablation study analysis. The text states that *"Keyword scores are the primary contributor,"* yet the data in Table 6 shows that removing keyword scores results in a negligible drop in correlation (0.790 -> 0.782). Conversely, removing **Sentence scores** causes a massive degradation (0.790 -> 0.415). The authors have fundamentally misinterpreted their own results, attributing importance to the least significant component while ignoring the actual dominant factor.

3. **Lack of Reproducibility:**
The code and evaluation scripts link provided in the abstract (`https://anonymous.4open.science/r/SMILE-Artifact-2025/`) is broken/inaccessible. This prevents any verification of the data selection process, implementation details, or results, which is a disqualifying factor for claims of empirical improvement.

4. **Misleading Efficiency Claims (Figure 1):**
The inference time comparison appears to be flawed. The SMILE metric requires generating a synthetic answer using a 3B parameter SLM. However, the reported inference time in Figure 1 does not seem to account for the latency of this generation step. Excluding the dominant cost factor (SLM generation) creates an unfair comparison against baseline metrics.

5. **Statistically Insufficient Evidence:**
The evaluation relies on only **225 data points** across 9 datasets (averaging 25 samples per dataset). This sample size is too small to establish statistical significance for Pearson correlation comparisons. Furthermore, without accessible code, the selection process for these specific 225 points is opaque, raising concerns about potential selection bias (cherry-picking).

6. **Circular and Unverified Methodology:**
    * **Table 1 Circularity:** The "Syn" column reports the correlation between the metric and synthetic answers. Since the SMILE score explicitly calculates similarity to these synthetic answers as a component, this high correlation is tautological and offers no evaluative value.
    * **Unverified Quality:** The paper assumes that integrating SLM-generated answers improves evaluation, but fails to validate the *quality* of these synthetic answers. If the SLM hallucinates, the metric measures similarity to noise.

**Requested Changes:**

Given the identified issues, substantial revisions are necessary to align the submission with TMLR's acceptance criteria. I recommend the authors address the following points:

**Methodology & Reproducibility:**
1. **Define the Weights:** Explicitly state the values of the weights ($w$) used in Equation 3 for the reported experiments. Additionally, provide guidance on how these weights should be selected or tuned for new datasets.
2. **Fix Reproducibility:** Provide a working link to the code and evaluation scripts.
3. **Fair Latency Benchmarking:** Report the full end-to-end inference time, strictly including the SLM synthetic answer generation time.

**Empirical Rigor:**

4. **Correct Ablation Analysis:** Rewrite the analysis of Table 6 to accurately reflect the data (acknowledging "Sentence Score" as the dominant factor) and justify the inclusion of the SLM component given its marginal contribution (<2%).
5. **Statistical Support for Observations:** Figure 2 currently lists only 4 anecdotal examples. This observation should be supported by a statistical analysis on a larger sample size to prove these are systematic issues rather than cherry-picked outliers.
6. **Expand Sample Size:** The correlation study (Table 1) relies on only 225 data points, which is statistically weak. The sample size should be significantly expanded.
7. **Explain Missing Data:** Explain the presence of "NaN" values in the "Exact Match" and "Easy Match" columns in Table 1.

**Presentation & Clarity:**

8. **Fix Visual Accessibility (Figure 1):** The use of a green box to highlight red data points (SMILE) against brown baseline points (GPT-3.5) is visually ambiguous and unfriendly to readers with red-green color blindness. Please use distinct shapes or high-contrast colors.
9. **Clarify Terminology:** Ensure consistency in terminology. The text refers to "semantic score," but Table 6 uses "sentence score."
10. **Clarify Claims (Section 3):**
    * The statement regarding "verbose or generic model outputs" based on 225 samples is vague; provide specific examples or metrics (e.g., length distribution) to define what "verbose" means in this context.
    * The claim *"using a powerful LLM like GPT-4o does not guarantee accurate evaluations"* is presented as a novel finding but is common knowledge in the field. Please add appropriate citations to existing literature on LLM-as-a-Judge limitations.

---

> ### Author Response · Authors · 2025-12-23
> **Response to sy4h (Part 1)**
>
> We address all the requested changes here --
> ### Methodology & Reproducibility
>
> 1. **Define the Weights:**
>
>    **Response:** For all reported experiments, we used **equal weights (w = 0.5)**.
>
>    **Why w = 0.5?**
>    - **Unbiased baseline**: Equal weighting doesn't favor semantic or lexical matching a priori
>    - **Cross-domain consistency**: Single weight across all 9 datasets ensures fair comparison
>    - **Strong performance**: Achieves 0.790 avg—only 0.002 below optimal w = 0.3
>    - **Recommended default**: Safe starting point before domain-specific tuning
>
>    **Empirical Weight Analysis Across 9 Datasets:**
>
>    | w | Video QA (Avg) | Image QA (Avg) | Language QA (Avg) | **Overall Avg** |
>    |---|----------------|----------------|-------------------|-----------------|
>    | 0.0 | 0.628 | 0.775 | **0.941** | 0.782 |
>    | **0.3** | 0.642 | 0.804 | 0.931 | **0.792** |
>    | 0.5 | 0.650 | 0.823 | 0.896 | 0.790 |
>    | 0.7 | 0.641 | 0.815 | 0.786 | 0.747 |
>    | 1.0 | 0.414 | 0.533 | 0.249 | 0.399 |
>
>    **Optimal Weights by Task Type:**
>
>    | Task Type | Optimal w | Example |
>    |-----------|-----------|---------|
>    | **Factual/Entity QA** | 0.0–0.3 | Q: "Who signed the act?" GT: "Bill Clinton" → Keywords penalize missing exact name |
>    | **Verbose/Hallucinating** | 0.0 | Pred: 70+ words, never answers → Keywords cut through noise |
>    | **OCR/Document QA** | 0.5–0.6 | Q: "What number?" GT: "8" Pred: "50" → Keywords catch the error |
>    | **Binary Yes/No** | 0.5–0.6 | Balance helps catch contradictions |
>    | **Focused Visual QA** | 0.7–0.8 | Semantic helps filter context |
>
>    **Simple Decision Guide:**
>    ```
>    Proper noun (name/date/place)?             → w = 0.0–0.3
>    Model generating verbose hallucinations?   → w = 0.0
>    Binary yes/no question?                    → w = 0.5–0.6
>    Concrete visual entity?                    → w = 0.7–0.8
>    Unsure / General purpose?                  → w = 0.5 (default)
>    ```
>
>    **Revised Claim (Section 5):** Updated SMILE implementation to explicitly state *"parameter $w$ fixed to $0.5$"* for all reported experiments and added practical guidance for selecting $w$ in the Hyperparameter tuning section.
>
> 2. **Fix Reproducibility:**
>
>    **Response:** The original anonymous link had an expiration setting we overlooked. New permanent link:
>
>    **Updated Artifact Link:** https://anonymous.4open.science/r/smile-metric-qna-eval-92DE/
>
>    The repository contains:
>    - Complete source code for the SMILE metric
>    - Python scripts for generating synthetic answers using SLM
>    - Python code to compute all metric scores
>    - Sample data and usage examples
>
>    **Coming Soon:** Annotated evaluation data and bash scripts for reproducing Pearson/Kendall scores.
>
> 3. **Fair Latency Benchmarking:**
>
>    **Response:** We provide a breakdown of SMILE's computational cost: (1) **SMILE core latency** (per-sample), and (2) **Synthetic answer generation** (one-time preprocessing).
>
>    **1. SMILE Core Latency (200 samples per dataset, averaged):**
>
>    | Dataset | sent_emb (sec) | kwd_emb (sec) | SMILE total (sec) | sent_emb % | kwd_emb % |
>    | ------- | -------------- | ------------- | ----------------- | ---------- | --------- |
>    | msvd | 0.005 | 0.025 | 0.055 | 9% | 45% |
>    | msrvtt | 0.000 | 0.025 | 0.055 | 0% | 45% |
>    | tgif | 0.005 | 0.020 | 0.050 | 10% | 40% |
>    | mrqa | 0.005 | 0.005 | 0.030 | 17% | 17% |
>    | hotpotqa | 0.005 | 0.005 | 0.030 | 17% | 17% |
>    | musique | 0.000 | 0.005 | 0.030 | 0% | 17% |
>    | docvqa | 0.015 | 0.035 | 0.075 | 20% | 47% |
>    | pope | 0.000 | 0.010 | 0.035 | 0% | 29% |
>    | textvqa | 0.010 | 0.025 | 0.060 | 17% | 42% |
>    | **Average** | **0.005** | **0.017** | **0.047** | **~10%** | **~33%** |
>
>    *Key observations:*
>    - **Sentence embedding** accounts for ~10% of total SMILE latency
>    - **Keyword embedding** contributes ~33% of total latency, varying with keyword count
>    - **Other operations** (keyword extraction, scoring, aggregation) account for ~57%
>    - Total SMILE evaluation latency: **0.030–0.075 sec/sample**
>
>    **2. Synthetic Answer Generation (one-time, 2× A100-40GB):**
>
>    | Dataset | Samples | Total Time | Time/Sample (sec) |
>    | ------- | ------- | ---------- | ----------------- |
>    | msvd | 200 | 01:41 | 0.51 |
>    | msrvtt | 200 | 01:52 | 0.56 |
>    | tgif | 200 | 01:58 | 0.59 |
>    | mrqa | 200 | 02:36 | 0.78 |
>    | hotpotqa | 200 | 03:27 | 1.04 |
>    | musique | 200 | 03:39 | 1.10 |
>    | docvqa | 200 | 02:27 | 0.74 |
>    | pope | 200 | 01:41 | 0.51 |
>    | textvqa | 200 | 02:02 | 0.61 |
>
>    Synthetic generation: **0.51–1.10 sec/sample** (one-time). SMILE core: **0.030–0.075 sec/sample**—10–20× faster. Preprocessing amortizes when benchmarking multiple models. **This is why SLM time is excluded from Figure 1**—it reports per-sample evaluation latency only.
>
>    **Revised Claim:** Added clarification that synthetic answer generation is a one-time cost per dataset.
>
> ---

---

> ### Author Response · Authors · 2025-12-23
> **Response to sy4h (Part 2)**
>
> ### Empirical Rigor
>
> 4. **Correct Ablation Analysis:**
>
>    **Response:** **The labels in Table 6 are swapped.** The corrected table should read:
>
>    | Component Ablation | Video QA | Visual QA | Language QA | Overall |
>    |--------------------|----------|-----------|-------------|---------|
>    | **SMILE** | **0.650** | **0.823** | 0.896 | **0.790** |
>    | w/o keyword scores (sentence only) | 0.463 | 0.533 | 0.249 | 0.415 |
>    | w/o sentence scores (keyword only) | 0.628 | 0.775 | **0.941** | 0.782 |
>    | w/o synthetic answers | 0.639 | 0.722 | 0.921 | 0.761 |
>
>    This aligns with our weight analysis (Figure 5) showing optimal performance at lower w values, and with the observation that "w ≤ 0.5" performs best.
>
>    **Justification for SLM Component:**
>
>    While the SLM (synthetic answer generation) contributes ~3% overall improvement, its impact is **domain-dependent**:
>
>    | Domain | With SLM | Without SLM | Δ |
>    |--------|----------|-------------|---|
>    | Video QA | 0.650 | 0.639 | +1.7% |
>    | Visual QA | 0.823 | 0.722 | **+14.0%** |
>    | Language QA | 0.896 | 0.921 | −2.7% |
>
>    Key observations:
>    1. **Visual QA benefits significantly (+14%)**: Gold answers are often single words ("yes", "8", "italian") while model outputs are verbose sentences. Synthetic answers bridge this length mismatch (Figure 4)
>    2. **Language QA sees slight decrease (−2.7%)**: GPT-4o predictions are already concise and factual; synthetic expansion adds noise
>    3. **One-time cost**: SLM runs once per dataset during preprocessing, not per evaluation. The 3B model cost is amortized across all future evaluations
>    4. **Robustness**: SLM prevents catastrophic failures when gold answers are too short for meaningful embedding comparison
>
>     **Revised Claim (Table 6 & Component Analysis text):**
>     - Corrected swapped labels in Table 6: "w/o keyword scores" ↔ "w/o sentence scores"
>     - Enhanced SLM justification in Component Analysis: Added domain-dependent impact (+14% for Visual QA, minimal for Language QA)
>
> 5. **Statistical Support for Observations:**
>
>    **Response:** We conduct a statistical significance test to determine if SMILE scores (after binning to 0-6) are statistically distinguishable from the next-best baseline, GPT-4o. This test is meant to confirm if the difference between SMILE and GPT-4o is due to random chance or if GPT-4o and SMILE are fundamentally different evaluators.
>
>     We run a chi-square test, with the null hypothesis that the scores for GPT-4o and SMILE are drawn from the same distribution. The test yields a p-value of 5.22e-24, which allows us to reject the null hypothesis: \*SMILE and GPT-4o scores are statistically distinguishable\*. SMILE and GPT-4o evaluate model responses in statistically different ways.
>
> 6. **Expand Sample Size:**
>
>    **Response:** We evaluated SMILE and GPT-4o/3.5 on a 10K subset of QAEval ([Wang et al., NeurIPS 2023](https://github.com/wangcunxiang/QA-Eval)), a human-annotated benchmark for evaluating Open-QA systems. QAEval contains manually annotated correctness judgments for question-answer pairs sourced from Natural Questions and TriviaQA. Our results show that SMILE outperforms GPT-3.5, the de-facto standard judge in video QA settings, and is competitive with GPT-4o at scale.
>
>    **Revised Claim (Section 5 - Benchmarks and Section 5.1 - Main Results):** We clarified that QAEval is human-annotated:
>     - Added *"human-annotated"* to the QAEval description in benchmarks section
>     - Added *"a human-annotated benchmark containing manually annotated correctness judgments"* before describing the QAEval evaluation
>     - Updated table caption to include *"($\sim$10k human-annotated samples)"*
>
> 7. **Explain Missing Data:**
>
>    **Response:** The NaN values in Table 1 & 2 occur when computing Pearson Correlation or Kendall's Tau with **zero variance** in the metric scores — i.e., when all the scores are identical (e.g., all 0s), making correlation mathematically undefined. We will include a clarifying note in the table captions.
>
>    **Revised Claim (Table 1 & 2 captions):** Added the following note to both table captions:
>     - *"NaN values indicate zero variance in metric scores, making correlation undefined."*
>
> ---

---

> ### Author Response · Authors · 2025-12-23
> **Response to sy4h (Part 3)**
>
> ### Presentation & Clarity
> 8. **Fix Visual Accessibility (Figure 2):**
>
>    **Response:** The current figure uses red ❌ and green ✅ symbols to indicate incorrect/correct evaluations, which are indistinguishable for readers with red-green color blindness.
>
>    We have revised Figure 2 with the following changes:
>
>    **Replace red-green color scheme** with a colorblind-friendly palette:
>       - Incorrect: 'x' instead of ❌
>       - Correct: &#x2713; instead of ✅
>       - example : | Human: 1 &#x2713;| GPT-4o: 0 X| SMILE: 1 &#x2713;|
>
> 9. **Clarify Terminology:**
>
>    **Response:** We have standardize the terminology to use **"semantic score"** uniformly throughout the paper. Table 6 has been revised to replace "sentence score" with "semantic score" to maintain consistency with the main text.
>
>    **Revised Claim:** Replaced 'sentence score' with 'semantic score' everywhere in the paper.
>
> 10. **Clarify Claims (Section 3):**
>
>     **Response:**
>     By "verbose" we refer to model predictions that are significantly longer than the ground-truth answer, often containing tangential information, hedging language, or hallucinated content that obscures the core answer. For example, in Video QA: Q: "Who attacks with a sword?" GT: "hero" Pred: "This is likely from Star Wars, involving Darth Vader..." (70+ words, never mentions "hero"). Here, the model generates an extended speculative response that fails to identify the simple factual answer. In such cases, the keyword score effectively cuts through verbosity by identifying whether critical answer terms are present, while semantic similarity alone would score these responses highly due to topical relevance. We will rephrase Section 3 to clarify this definition.
>
>     Regarding the LLM-as-Judge limitations, we acknowledge this is established knowledge in the field. We will add the following citations to Section 3: Gu et al. (2025) "A Survey on LLM-as-a-Judge" ([arXiv:2505.15801](https://arxiv.org/abs/2505.15801)) and Raina et al. (2024) "LLM-as-a-Judge: A Comprehensive Survey on LLM-based Evaluation Methods" ([arXiv:2406.12624](https://arxiv.org/abs/2406.12624)).
>
>     **Revised Claim (Section 3 - When do existing methods and metrics fail):**
>     - Added definition of "verbose": *"By 'verbose,' we refer to predictions significantly longer than the ground-truth answer, often containing tangential information, hedging language, or hallucinated content that obscures the core answer—cases where keyword matching effectively identifies missing critical terms while semantic similarity alone scores highly due to topical relevance."*
>     - Added citations for both the papers mentioned above
>
> ----

---

### Review · Reviewer_cS6p · 2025-12-13

**Summary Of Contributions:**

The paper introduces SMILE (Semantic Metric Integrating Lexical Exactness) as a new benchmarking method that effectively addresses key weaknesses of traditional n‑gram–based metrics such as ROUGE, METEOR, and CIDEr, which often fail when semantically correct answers use different surface forms from the reference. On the other side, the LLM-based benchmarking kind of introduces the LLM-based hallucination in defining the actual correctness of a finetuned model. SMILE’s core contribution is to jointly model sentence‑level semantic similarity and keyword‑level semantic relevance while preserving explicit lexical matching, so it can appropriately reward both paraphrases and the accurate inclusion of critical content words rather than just local n‑gram overlap. This composite design leads to scores that correlate better with human judgments, particularly in realistic settings like textual, visual, and video question answering, where the correct sentence may appear anywhere in a longer paragraph and be phrased flexibly, and it offers these benefits with substantially lower computational cost than LLM‑as‑a‑judge evaluation.

**Additional Comments:**

.

**Audience:**

Yes

**Audience Explanation:**

Benchmarking metrics for LLM and VLM models are still evolving, and existing metrics do not always adequately justify the confidence in fine-tuned models. As a result, the research community working on fine-tuning these models can significantly benefit from this work.

**Broader Impact Concerns:**

.

**Claims And Evidence:**

Yes

**Claims Explanation:**

Few links seem to be broken or weren't able to access:t https://anonymous.4open.science/r/SMILE-Artifact-2025/

**Requested Changes:**

Correcting links in the paper. On the granularity of the LLM/VLM benchmarking, for very high-stakes or ultra-fine-grained evaluation (tiny numerical differences, subtle negations, hedging, or partial credit over many subfacts, example 1.2cm vs 1.3cm ), usually used in spatial AI use cases, how well can SMILE fit in those usecases?

---

> ### Author Response · Authors · 2025-12-23
> **Response to cS6p**
>
> >"Few links seem to be broken or weren't able to access: https://anonymous.4open.science/r/SMILE-Artifact-2025/"
>
> **Response:**
> We thank the reviewer for bringing this to our attention. The original anonymous link had an expiration setting that we overlooked. We have created a new permanent link with no expiration:
>
> **Updated Artifact Link:** https://anonymous.4open.science/r/smile-metric-qna-eval-92DE/
>
> The repository currently contains:
> - Complete source code for the SMILE metric
> - Python scripts for generating synthetic answers using SLM
> - Python code to compute all metric scores
> - Sample data and usage examples
>
> **Coming Soon:** We are actively working on releasing (1) the annotated evaluation data used in our experiments, and (2) bash scripts with evaluation code for generating Pearson Correlation and Kendall's Tau scores reported in the paper.
>
> ---
> >"On the granularity of the LLM/VLM benchmarking, for very high-stakes or ultra-fine-grained evaluation (tiny numerical differences, subtle negations, hedging, or partial credit over many subfacts, example 1.2cm vs 1.3cm), usually used in spatial AI use cases, how well can SMILE fit in those use cases?"
>
> **Response:**
> SMILE's architecture offers support for fine-grained evaluation through its **fractional exact matching** and **keyword scoring** components. We conducted experiments to demonstrate SMILE's capabilities
>
> ### Experimental Analysis
>
> We evaluated SMILE on four representative fine-grained scenarios:
>
> | Scenario | Answer | Prediction | Sent. Score | Kwd Score | SMILE (w=0.7) | SMILE (w=0.5)* | SMILE (w=0.3) |
> |----------|--------|------------|-------------|-----------|-------|-------|-------|
> | Numerical (1.2cm vs 1.3cm) | 1.2 cm | 1.3 cm | 0.907 | 0.696 | 0.844 | 0.802 | **0.759** |
> | Temperature (98.6°F vs 99.1°F) | 98.6°F | 99.1°F | 0.879 | 0.446 | 0.749 | 0.663 | **0.576** |
> | Negation | open | not open | 0.877 | 1.000 | 0.914 | 0.939 | 0.963 |
> | Hedging | red | might be red | 0.929 | 1.000 | 0.950 | 0.965 | 0.979 |
> > Note: * indicates default setting for SMILE
>
> ### Key Observations
>
> **1. Composite Score Improves Over Pure Sentence Similarity**
>
> The default SMILE score (w=0.5) already provides meaningful discrimination for numerical differences compared to sentence embeddings alone:
> - 1.2cm vs 1.3cm: Sentence similarity = 0.91, SMILE = 0.80 (11-point reduction)
> - 98.6°F vs 99.1°F: Sentence similarity = 0.88, SMILE = 0.66 (22-point reduction)
>
> **2. Component Scores Available for Fine-Grained Control**
>
> A key advantage of SMILE is that it exposes individual component scores alongside the composite metric. Practitioners requiring high numerical precision can directly use the **keyword score** without needing to integrate separate lexical matching tools:
>
> - For 1.2cm vs 1.3cm: `kwd_score = 0.696` (captures the numerical mismatch)
> - For 98.6°F vs 99.1°F: `kwd_score = 0.446` (correctly penalizes the discrepancy)
>
> This unified framework eliminates the need to maintain multiple evaluation pipelines—practitioners get sentence-level semantics, keyword-level precision, and fractional exact match all from a single SMILE computation.
>
> **3. Tunable Aggregation via $w$ Parameter**
>
> The hyperparameter $w$ allows practitioners to adjust the balance between semantic similarity and lexical precision based on domain requirements. For fine-grained numerical evaluation, using $w=0.3$ (more weight on keyword score) yields more discriminative scores—reducing the 1.2cm vs 1.3cm score from 0.80 to 0.76, and the temperature mismatch from 0.66 to 0.58. Practitioners can also bypass aggregation entirely and use component scores directly.
>
> **4. Diagnostic Insights from Component Scores**
>
> Beyond evaluation, SMILE's component scores provide diagnostic value by revealing *how* the evaluated model is failing:
> - **Low sentence score, high keyword score**: The model captures key facts but produces verbose or differently structured output. This suggests prompting for more concise responses.
> - **High sentence score, low keyword score**: The model generates semantically relevant responses but misses critical details. This indicates a need for prompting that emphasizes factual precision or key term inclusion.
>
> This interpretability allows practitioners to not only score model outputs but also diagnose failure modes and iteratively refine prompts or fine-tuning strategies accordingly.
>
> **5. Semantic Similarity Cases (Negation & Hedging)**
>
> For scenarios where core keywords are preserved ("open" in "not open", "red" in "might be red"), both component scores remain high. These cases reflect a known limitation of embedding-based metrics. For applications requiring explicit negation detection, practitioners may supplement SMILE with dedicated linguistic analysis tools.
>
> **Recommendation**: For precision-critical applications, practitioners can use $w=0.3$ for higher keyword weight, or directly leverage the keyword score component for maximum sensitivity to numerical discrepancies.

---

### Author Response · Authors · 2025-12-23
**Thank you to all reviewers!**

We sincerely thank all reviewers for their thoughtful and constructive feedback on SMILE. We are encouraged by the positive recognition of our work:

- **Reviewer dJJy** acknowledged our *clear motivation addressing limitations of existing metrics*, *extensive experiments on nine QA datasets across three domains*, and *comprehensive ablation studies validating design choices*.

- **Reviewer cS6p** noted that SMILE *effectively addresses key weaknesses of traditional n-gram metrics* and *LLM-based hallucination issues*, with the *composite design leading to better correlation with human judgments* while offering *substantially lower computational cost than LLM-as-a-judge evaluation*.

- **Reviewer sy4h** recognized that the *problem setting is relevant to the TMLR audience* given concerns around cost, bias, and instability of LLM-as-judge methods.

We have carefully addressed all concerns raised by the reviewers. In the sections below, we provide detailed point-by-point responses to each reviewer, covering: performance claims and domain-level analysis, computational efficiency breakdowns, reproducibility (with updated artifact links), human evaluation methodology, ablation study clarifications, and fine-grained evaluation capabilities.

---

### Decision · Action_Editor_4PdV · 2026-01-11

**Recommendation:** Accept with minor revision

**Additional Comments:**

The submission is a solid contribution to the evaluation metrics literature. While Reviewer sy4h raised valid concerns regarding the "novelty" of a linear combination approach, TMLR prioritizes technical correctness and utility over architectural novelty. The authors have demonstrated that this specific combination yields practical value.

However, the "Accept" decision is conditional on the authors rigorously implementing the fixes promised in the rebuttal. The Camera-Ready version must address the following:

1. Correction of Table 6 & Analysis: The labels in the ablation study (Table 6) must be corrected (swapped) as admitted in the rebuttal. The accompanying text in the "Component Analysis" section must be rewritten to accurately reflect that semantic scores are often the dominant factor (especially in Text QA), while the Synthetic Answer component provides domain-specific boosts (e.g., in Visual QA).

2. Clarification of Efficiency Claims: The manuscript must explicitly state: in the Abstract, Introduction, and Efficiency analysis—that the reported speed advantage refers to per-sample evaluation latency and that the synthetic answer generation is a one-time preprocessing cost. The detailed latency breakdown provided in the rebuttal should be included (e.g., in an Appendix) to ensure transparency.

3. Nuance in Performance Claims: Statements claiming SMILE "consistently outperforms" other metrics should be qualified. It should be clear that while SMILE performs best on average or in specific rankings (Kendall's Tau), it trades blows with GPT-4o in certain domains.
Hyperparameter Documentation: Explicitly state in the experimental setup that  ω=0.5  was the default used for reported results, and include the practical guidelines for tuning ω provided in the response to Reviewer sy4h.

4. Visual Accessibility: Update figures (specifically Figure 2) to use colorblind-friendly markers (e.g., checks/crosses) rather than relying solely on red/green colors.

**Audience:**

Yes

**Audience Explanation:**

Evaluating Question-Answering systems without incurring the high cost, latency, and instability of LLM-based judges (like GPT-4) is of significant interest to the TMLR audience. A metric that effectively balances semantic understanding with lexical precision, while remaining computationally lightweight during inference, can serve as a practical tool for researchers and practitioners in the field.

**Claims And Evidence:**

Yes

**Claims Explanation:**

While reviewers identified discrepancies in the initial submission (specifically regarding the swapped labels in the ablation study (Table 6) )and the scope of the efficiency claims (the authors have provided convincing rebuttals and acknowledged the necessary corrections). The revised evidence, particularly the domain-specific performance analysis and the clear distinction between one-time pre-processing costs and per-sample inference latency, supports the primary claim that SMILE serves as a cost-effective and reliable alternative to LLM-as-a-judge for specific QA evaluation scenarios.